# Precise phylogenetic analysis of microbial isolates and genomes from metagenomes using PhyloPhlAn 3.0

Francesco Asnicar [1], Andrew Maltez Thomas [1], Francesco Beghini [1], Claudia Mengoni[1], Serena Manara [1], Paolo Manghi[1], Qiyun Zhu[2], Mattia Bolzan [1,9], Fabio Cumbo [1], Uyen May[3], Jon G. Sanders [2,12], Moreno Zolfo [1], Evguenia Kopylova[2,11], Edoardo Pasolli[1,10], Rob Knight [2,4,5,6], Siavash Mirarab [3], Curtis Huttenhower [7,8] & Nicola Segata [1✉]

Microbial genomes are available at an ever-increasing pace, as cultivation and sequencing become cheaper and obtaining metagenome-assembled genomes (MAGs) becomes more effective. Phylogenetic placement methods to contextualize hundreds of thousands of genomes must thus be efficiently scalable and sensitive from closely related strains to divergent phyla. We present PhyloPhlAn 3.0, an accurate, rapid, and easy-to-use method for large-scale microbial genome characterization and phylogenetic analysis at multiple levels of resolution. PhyloPhlAn 3.0 can assign genomes from isolate sequencing or MAGs to species-level genome bins built from >230,000 publically available sequences. For individual clades of interest, it reconstructs strain-level phylogenies from among the closest species using clade-specific maximally informative markers. At the other extreme of resolution, it scales to large phylogenies comprising >17,000 microbial species. Examples including *Staphylococcus aureus* isolates, gut metagenomes, and meta-analyses demonstrate the ability of PhyloPhlAn 3.0 to support genomic and metagenomic analyses.

[1] Department CIBIO, University of Trento, Trento, Italy. [2] Department of Pediatrics, University of California San Diego, La Jolla, CA, USA. [3] Department of Electrical and Computer Engineering, University of California San Diego, La Jolla, CA, USA. [4] Department of Computer Science and Engineering, University of California San Diego, La Jolla, CA, USA. [5] Center for Microbiome Innovation, University of California San Diego, La Jolla, CA, USA. [6] Department of Bioengineering, University of California San Diego, La Jolla, CA, USA. [7] Department of Biostatistics, Harvard T. H. Chan School of Public Health, Boston, MA, USA. [8] The Broad Institute of MIT and Harvard, Cambridge, MA, USA. [9]Present address: PreBiomics s.r.l, Trento, Italy. [10]Present address: Department of Agricultural Sciences, University of Naples Federico II, Portici, Italy. [11]Present address: Clarity Genomics BVBA, Sint-Michielskaai 34, 2000, Antwerpen, Belgium. [12]Present address: Cornell Institute for Host-Microbe Interaction and Disease, Cornell University, Ithaca, NY, USA. ✉email: nicola.segata@unitn.it

Genomes from isolate sequencing, metagenomic assembly, and single-cell sequencing are being generated at an increasing pace, and they are all correspondingly increasingly available through public resources. This provides invaluable insights into the overall characterization of microbial diversity affecting the human body and the planet. Phylogenetic and corresponding taxonomic characterization is crucial in microbial genomics, for contextualizing genomes without prior phenotypic information, and for determining their genetic novelty and genotype-phenotype relationships. At the largest scale, reconstructing a complete microbial tree-of-life is fundamental in understanding evolutionary relationships in any context, and in microbial community studies such a reference can be a crucial link between novel sequences and health or environmentally relevant microbes. Regardless of the scale, many current microbial genomic tasks thus include the need to place newly sequenced genomes and metagenomic assembled genomes into the microbial taxonomy and phylogenetically characterize them with respect to the closest relatives. With such a volume of microbial genomes generated at a wide range of qualities and completeness, however, there are no scalable phylogenetic methods that can easily tackle these challenges for investigators studying genomes and metagenomes.

Many methods exist for more targeted microbial genome and metagenome phylogenetics. These, include the first implementation of PhyloPhlAn[1], PhyloSift[2], ezTree[3], GToTree[4], and AMPHORA[5], among many others for more general genome- and gene-based phylogenetics[6,7]. Most of these methods are limited in at least one way that prevents their ease of use to link newly sequenced genomes, or metagenomic assemblies, into the tremendous space of already characterized microbial phylogenies. None, for example, allow different genomic regions to be selected to achieve optimal resolution in differing clades. This both degrades performance for some clades and prohibits the same methods from being used for strain-level versus phylum-level placement. None leverage the complete set of >100,000 publicly available microbial genomes and and of >200,000 metagenome-assembled genomes (MAGs) from >10,000 metagenomes, and while GToTree automatically retrieves reference genomes from public resources, it does not provide access to MAGs or phylogenetic markers for species-level clades. While computational methods for genome assembly of isolate sequencing and for

quantitative analysis of known features of metagenomic data are now mature and well standardized, comparably convenient and automatic tools for downstream phylogenetic and taxonomic assessment of MAGs and microbial isolate genomes are instead lacking and limiting microbial genomic analyses.

These end-to-end phylogenetic solutions should also be differentiated from algorithms and implementations for individual steps of genome placement (e.g., pplacer[8] and SEPP[9]) and taxonomic assessment. Examples include algorithms for multiple-sequence alignment (MSA) like MUSCLE[10], MAFFT[11], T-Coffee[12], OPAL[13], PASTA[14], and UPP[15] and phylogenetic reconstruction like FastTree[16,17], RAxML[6], ASTRAL[18–20], ASTRID[21], and IQ-TREE[22]. Each tool can be separately and sequentially applied providing full step-by-step control on the whole phylogenetic analysis, but doing so requires substantial expertize not only in identifying the right targets, parameters, and steps for computational phylogenetics, but also in understanding how such tools should be interfaced one with the other.

Separate and human-supervised execution of these steps is also impractical when individual studies generate thousands of microbial genomes, or when massive numbers of genomes are retrieved and analyzed in combination. Very efficient algorithms have been proposed, including those based on typing only a few representative marker genes, such as multilocus sequence typing (MLST) approach[23] or on species-level core genes[24]. Computational MLST, for instance, can operate rapidly using as few as five to ten loci for each species. However, this comes at the cost of greatly reduced accuracy of phylogenetic placement. Pangenome-based profiling like Roary[24] is instead very accurate for phylogenetic modeling at species level but cannot be generalized to higher-level clades. Strain-resolved phylogenies integrating thousands of reference genomes from diverse species—or at least those most closely related to new sequences of interest—result in a more accurate characterization of microbes' population structure and characteristics, while also more accurately guiding taxonomy. Whole-genome large-scale microbial phylogenies, particularly robust to partial assemblies and able to integrate existing genomes and metagenomic assemblies, are thus an open computational challenge.

We thus present here PhyloPhlAn 3.0, a fully automatic, end-to-end phylogenetic analysis framework for contextualization and characterization of newly assembled microbial isolates and

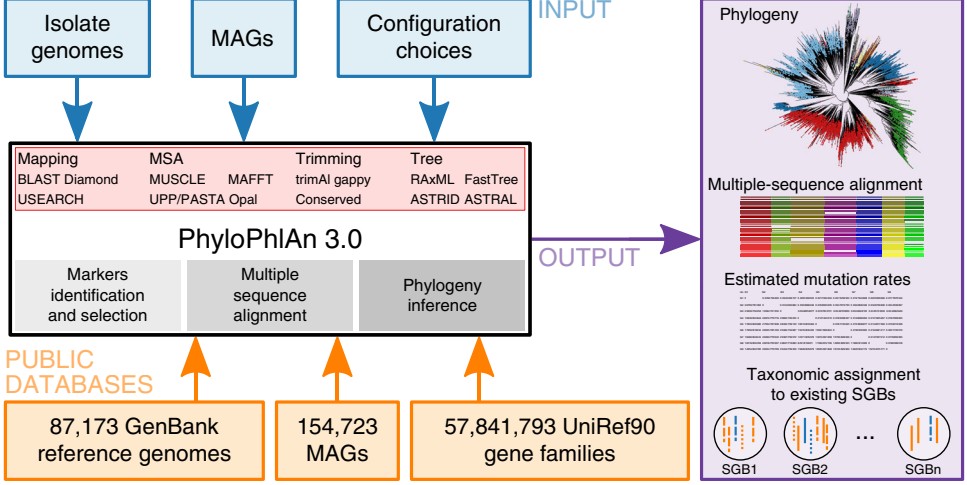

**Fig. 1 PhyloPhlAn 3.0 phylogenetically places microbial isolate or metagenomic assemblies.** PhyloPhlAn 3.0 provides strain-to-phylum level phylogenies built from newly generated microbial genomes (isolate or metagenomic assemblies) in the context of over 80,000 existing isolate genomes and 150,000 metagenomic assemblies. It automatically selects the most informative loci on a clade-specific basis, handles incomplete or fragmented assemblies, and can be configured to provide the resulting multiple-sequence alignment, estimated mutation rates (optionally), and phylogenetic tree.

metagenomes. PhyloPhlAn 3.0 can, as needed, retrieve and integrate hundreds of thousands of genomes from public resources, while also incorporating preprocessed information from tens of thousands of metagenomes. It automatically uses species-specific sets of core proteins, stably identified using UniRef90 gene families, to build accurate strain-level phylogenies, while also scaling to tens of thousands of genomes for inferring deep branching and very large size phylogenies. PhyloPhlAn 3.0 is both accurate at the strain and species level and fast when scaling to the whole set of available genomes. Compared to available alternatives such as the genome taxonomy database (GTDB)[25], PhyloPhlAn 3.0 is able to automatically perform taxonomic assignment of MAGs based on the NCBI taxonomy and to consider unnamed and uncharacterized species in the genomic contextualization task.

## Results

**Precise phylogenetic placement of genomes and metagenomes**. PhyloPhlAn 3.0 provides an easy-to-use and fully automatic method for accurate phylogenetic and taxonomic contextualization of microbial (meta)genomes (Fig. 1). The method can consider combined input sets of microbial genomes from isolate sequencing and of MAGs to produce phylogenies at multiple levels of resolution. Placement of input genomes and MAGs is performed by de novo reconstruction of the phylogeny. For highly resolved phylogenetic trees of related strains, PhyloPhlAn 3.0 uses species-specific core genes from the >18,000 sets of preselected UniRef90 gene families. Instead, for high-diversity genomes, it relies on the 400 most universal markers[1,26] with more aggressive alignment trimming options (see "Methods"). Multi-resolution phylogenetic reconstruction is also at the core of the approach to assign taxonomic labels from phylum to species level to input genomes or MAGs, which exploits >150,000 MAGs and >80,000 reference genomes integrated into the PhyloPhlAn 3.0 database. The pipeline thus integrates the large body of available whole-genome microbial data to phylogenetically contextualize input genomes by adopting several methodological advances depending on the characteristics and scale of the specific tasks (see "Methods"). PhyloPhlAn 3.0 is not bound to particular methodological choices for the internal steps: it allows users to choose among multiple tools for sequence mapping[27–29], MSA[10,11,13,30], post-processing of the alignments[31], with phylogenetic models ranging from maximum-likelihood methods applied on concatenated alignments[6,16,22] to gene tree approaches integrating the information of multiple distinct markers[18,21]. In addition to the phylogeny, rich outputs are provided ranging from taxonomic assignment to species-level genome bins (SGBs) to the raw multiple-sequence alignment and statistics of mutation rates for the genomes in the phylogeny. PhyloPhlAn 3.0 is a complete rewrite of the first PhyloPhlAn version[1] and a large number of features, as well as the ability to scale to tens of thousands of input genomes and MAGs, are unique to the new version (see Supplementary Data 1).

**Phylogenetic analysis of genomes within extant species**. The first way in which PhyloPhlAn 3.0 can place new genomes in the context of related reference organisms is when the target species is known. This may be the case, for example, in microbial genomics projects with tens, hundreds, or thousands of new isolates of the same organism (e.g., pneumococcus[32], meningococcus[33], and *Mycobacterium tuberculosis*[34]). In this case, the method uses sets of gene families specific to be maximally phylogenetically informative for the species under consideration. These markers are selected from the set 57.8 M gene families identified via UniRef90[35] to be core genes in at least one species

(i.e., present in all the genomes available for a species). When the input genomes are considered, the pipeline further screens the markers to keep only those that satisfy the core-gene definition considering the inputs, which is important for species with potentially low-quality markers as they comprise only a few representative genomes. Thus, for extant species, PhyloPhlAn 3.0 uses the largest possible number of sequence markers that are shared across all the genomes of the species of interest and that are automatically retrieved at runtime from the online supporting database thus avoiding any manual and time-consuming effort from the user.

As an illustrative example, we used PhyloPhlAn 3.0 to analyze 135 new *Staphylococcus aureus* isolate genomes we sequenced in the context of nosocomial infections[36] (Fig. 2a). We used a subset of 2127 total precomputed *S. aureus* core genes, 1658 of which were present in at least 99% of the genomes, which is one of the tunable parameters in the software package. We previously built a whole-genome phylogeny for these genomes by annotating assemblies via Prokka[37], computing a set of 1464 core genes (present in at least 99% of genomes) using Roary[24], manually editing the concatenated MSA to remove local misalignments, and finally using RAxML[6] to infer the associated phylogeny. While some minor differences do occur when comparing PhyloPhlAn 3.0's tree with the previous manually curated phylogeny (e.g., the placement of ST97 with respect to the closest sequence types [STs]), we found an overall correlation of 0.992 (Pearson's correlation) between normalized pairwise branch length distances from the two phylogenies (Fig. 2b) and 90.5% consistency between quartet distances (92% when considering one genome for each ST, Supplementary Fig. 2). For comparison, manual tree construction took approximately 131 CPU hours (including 3 h for gene annotation, 118 h for core-genome identification and MSA, and 10 h for tree construction) and several person-hours, while all the steps of PhyloPhlAn 3.0 ran in approximately 24 h on the same hardware (with no manual curation).

PhyloPhlAn 3.0 can further extend newly generated phylogenies to incorporate one or more existing reference genomes. To illustrate this, we used PhyloPhlAn 3.0 to add 1000 *S. aureus* reference genomes to the previous tree, automatically selected (from among 7259 total available genomes prioritized based on representativeness, see "Methods") and retrieved from GenBank[38], yielding a larger phylogeny of 1135 genomes (Supplementary Fig. 1). This tree is in close agreement with MLST typing[23], with very small intra-ST phylogenetic distance compared to inter-ST distances (0.0012 vs. 0.1256, respectively), and the resulting genetic context provides a clear interpretation for subspecies structure of the newly sequenced *S. aureus* isolates in the context of known species diversity (Fig. 2c).

**Robust taxonomy assignment for MAGs**. In addition to phylogenetic reconstruction, PhyloPhlAn 3.0 can assign a putative taxonomic label to uncharacterized genomes[40] if they can be confidently placed in well-labeled phylogenetic clades. Specifically, for each new genome, it identifies the closest SGB from the collection of known and newly defined candidate species spanning 154,723 MAGs and >80,000 isolate genomes[41]. These span 16,331 SGBs, of which 12,535 have a confident species label based on previous validations alleviating problems of NCBI taxonomic consistency because species labels are assigned to consistently clustered genomes by majority voting (see "Methods"). Following the definition of the SGBs, an input genome is assigned to an SGB (and its associated taxonomy, if any) if the Mash[42] average distance to the genomes in the bin is below 5% (see "Methods"), as this threshold has been suggested to be optimal for species

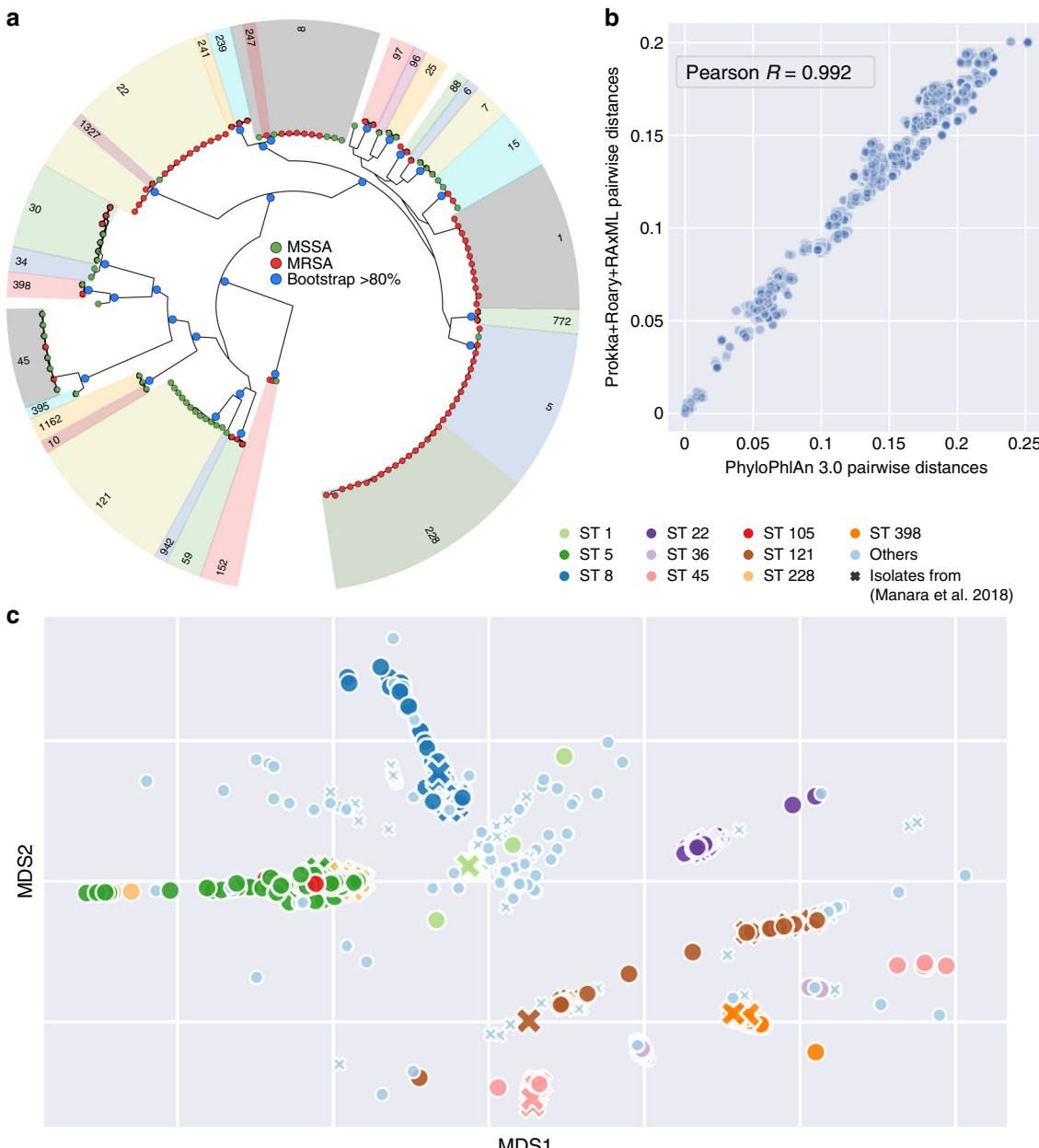

**Fig. 2 Accurate reconstruction of *Staphylococcus aureus* phylogenies using PhyloPhlAn 3.0. a** Phylogenetic tree of 135 *S. aureus* strains from a pediatric hospital[36] reconstructed by PhyloPhlAn 3.0 using 2127 automatically identified core genes (rendered by GraPhlAn[39] see Supplementary Fig. 2 for a full comparison). Green circles represent the methicillin-sensitive *S. aureus* (MSSA), while red circles represent methicillin-resistant *S. aureus* (MRSA). Blue circles internal to the phylogeny identify subtrees with bootstrap >80%. **b** Normalized phylogenetic distances in the PhyloPhlAn 3.0-reconstructed tree and in a manually curated phylogeny from ref. [36] highlighting strong consistency between the automated PhyloPhlAn 3.0 results and the curated tree (0.992 Pearson's correlation coefficient). **c** Multidimensional scaling ordination of pairwise phylogenetic distances from the tree integrating the 135 *S. aureus* isolates (crosses) with 1000 automatically selected *S. aureus* reference genomes (circles, Supplementary Fig. 1). The ten most prevalent sequence types (STs)[23] are highlighted in different colors.

definition[43,44]. If the input cannot be assigned to any SGB, then PhyloPhlAn 3.0 will report the set of closest SGBs (and their average genomic distances). If needed, this procedure is repeated for higher taxonomic clades with genus-level genome bins (GGBs, up to 15% genomic distance) and family-level genome bins (FGBs, up to 30% genomic distance, see "Methods")[41], ultimately providing a more comprehensive taxonomic context for the set of input genomes to guide downstream analyses and complement their phylogenetic placement. Validation on a set of 1520 isolate genomes from the gut microbiome[45] assigned an SGB to 1505 genomes (99%) demonstrating that the reference catalog of SGBs

covers very well the intestinal microbial diversity including 207 SGBs without a species name. The taxonomic labels inferred by PhyloPhlAn 3.0 were also very consistent (97.7%, Supplementary Data 2) with those assigned at species level in the original work highlighting the consistency of the automatic algorithm.

We used PhyloPhlAn 3.0 to taxonomically place a set of MAGs retrieved from a cohort of 50 rural Ethiopian individuals (see "Data availability") only used so far to characterize *Prevotella copri* strains[46], as these samples had not been used in the generation of SGBs and are likely to contain substantial unseen phylogenetic diversity. Overall, from the 369 medium- and high-quality input

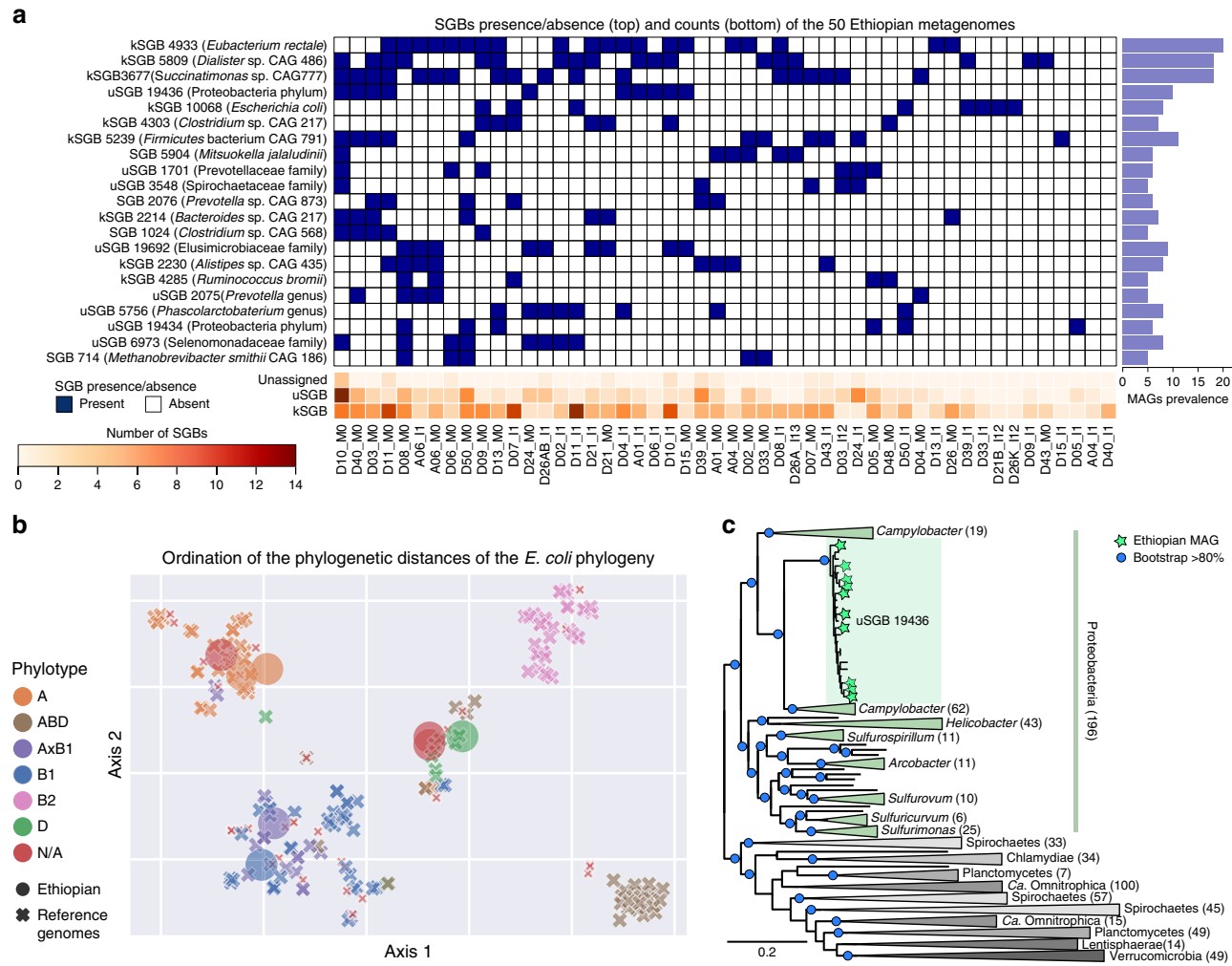

**Fig. 3 Phylogenetic analysis of MAGs from 50 rural Ethiopian metagenomes. a** Occurrence of the 20 most prevalent SGBs among 50 previously sequenced Ethiopian gut metagenomes highlights the presence of many previously identified but largely uncharacterized species-level genome bins (uSGBs) and the identification of few additional MAGs (unassigned) that are not recapitulated in any already defined SGB. The presence/absence profiles are clustered using average linkage with Euclidean distances. **b** Multidimensional scaling ordination using the t-SNE algorithm on phylogenetic distances from PhyloPhlAn 3.0's tree of eight Ethiopian *E. coli* MAGs (kSGB 10068) integrated with 200 automatically selected *E. coli* reference genomes using 3246 UniRef90 gene families for phylogenetic reconstruction. **c** PhyloPhlAn 3.0 phylogeny of Ethiopian MAGs assigned to uSGB ID 19436 including all reference genomes for the closest phyla (589 in total) according to the prokaryotes tree-of-life in Fig. 4. Phylogeny reconstruction used 400 universal markers selected by PhyloPhlAn 3.0 for deep-branching phylogenies. Portions of the tree collapsed are labeled and numbers in parentheses represent the number of genomes in the collapsed subtrees. Uncollapsed phylogeny is available in Supplementary Fig. 4.

MAGs (see "Methods"), PhyloPhlAn 3.0 provided an assignment to a total of 133 SGBs for 352 MAGs and a closest SGB indication for the remaining 17 MAGs. Twenty-one SGBs were detected in at least 5 samples (Fig. 3A), and the most prevalent SGBs were for *Eubacterium rectale* (ID 4933 found in 20 samples), an uncultured *Dialister* species (ID 5809, 18 samples) and an unnamed *Succinatimonas* species (ID 3677, 18 samples). While PhyloPhlAn 3.0 assigned an SGB to the majority of the genomes using the catalog of SGBs previously compiled through large-scale metagenomic assembly[41], a substantial number of these SGBs (39%) lacks taxonomic labels (uSGBs), further highlighting that microbiomes from rural communities contain many organisms that are still very poorly characterized. The few MAGs not assigned to known or unknown SGBs (17) belongs to candidate species that are specific to this cohort and for which none of the >154,000 MAGs from previous metagenomes are within a 5% whole-genome nucleotide similarity (and 14 at >10% genomic distance). This demonstrates that even with a very large reference set of genomes and metagenomes, cohort-specific microbes can be found, classified,

and phylogenetically profiled by the proposed approach. This study provides one example of how PhyloPhlAn 3.0 can automatically contextualize tens or hundreds of MAGs with taxonomy relative to characterized isolates or, when unavailable, using consistently cataloged microbial species from thousands of other metagenomes.

**Phylogenetic context for taxonomically unassigned genomes.** Since PhyloPhlAn 3.0 associates new genomes and MAGs with SGBs even when the latter do not contain previously characterized taxa, this can be used to automatically compare genomes and MAGs with hundreds or thousands of phylogenetically related genome sequences (Fig. 3). In the Ethiopian study, we focused on the prevalent human gut colonizer *Escherichia coli* (known SGB (kSGB) ID 10068), and on the most prevalent uSGB (ID 19436, 13 MAGs in total) for which the closest reference genomes belonged to the Proteobacteria phylum. Eight *E. coli* MAGs were constructed from the Ethiopian metagenomes, for which PhyloPhlAn 3.0 retrieved 200 reference genomes and 3246 UniRef90

families pre-calculated as core to the species (3099 of which were retained for phylogenetic reconstruction as they are consistently found in the eight input MAGs, Fig. 3b, Supplementary Fig. 3). This showed the eight Ethiopian input MAGs to be genetically heterogeneous, falling diversely among four different previously defined *E. coli* phylotypes (see "Methods") based on PhyloPhlAn 3.0-estimated phylogenetic distances (Fig. 3b, Supplementary Fig. 3). For half of the strains, the placement was confirmed by the phylogroup associated with the MLST types that could be inferred directly on the genomes[47], but the phylogenetic placement within the clustered phylotypes provides strong evidence for the assignment of the other four strains as well.

We used PhyloPhlAn 3.0 to place the uncharacterized uSGB 19436 in the context of other reference genomes and MAGs from the human microbiome[41] and of all the automatically retrieved species' representative genomes from the set of closest phyla that are: part of Proteobacteria (class Epsilonproteobacteria, non-monophyletic with the Proteobacteria phylum), Spirochetes, Chlamydiae, Planctomycetes, *Candidatus* Omintrophica, Lentisphaerae, and Verrucomicrobia, identified as being close to the Epsilonproteobacteria from the tree of life (Fig. 4). PhyloPhlAn 3.0 placed the expanded uSGB 19436 within several very divergent clades taxonomically assigned to the *Campylobacter* genus (Fig. 3c, Supplementary Fig. 4). The 812 publically available genomes in 108 SGBs assigned to distinct species of *Campylobacter*, reveal this genus to be extremely wide encompassing substantially more than 30% genetic distance (ANI analysis in Supplementary Fig. 4) which is a diversity usually characterizing whole classes or orders[41] suggests that this genus should be revised as also independently confirmed in other taxonomic reorganization efforts[25]. Although uSGB 19436 is rooted inside these divergent clades, its genetic divergence (Supplementary Fig. 4) is higher than typical family-level divergence and its phylogenetic distance is comparable to the distance between close phyla (Fig. 3c) thus supporting PhyloPhlAn 3.0's designation of a new species and genus. The new MAGs from the Ethiopian dataset also reinforce the observation that this phylogenetically divergent uSGB 19436 is specific of non-Westernized lifestyles as the previously reconstructed MAGs from this uSGB are all from populations with rural lifestyles in Madagascar[41], Peru[48], Tanzania[49], and Bangladesh[50]. This analysis thus highlights how PhyloPhlAn 3.0 can be used to expand the phylogenetic diversity of the human microbiome by the simple integration of MAGs from new cohorts in the already large set of microbial genome references considered by the method.

**PhyloPhlAn can scale to microbial tree-of-life phylogenies**. In addition to these small-to-medium examples of phylogenetic reconstruction for individual new genome sets, PhyloPhlAn 3.0 can scale to provide automatic placement of thousands of MAGs within the entire current microbial tree of life (Fig. 4). Specifically, we considered all high-quality microbial isolate genomes included in UniProt[51] (87,173 total), >154,000 MAGs from human-associated microbiomes[41], and ~8000 MAGs from primarily non-human environments[52]. These were dereplicated prior to PhyloPhlAn application to one representative per species by hierarchical clustering on genomic distances as estimated by Mash[42] with cluster cutoff at 5% intra-cluster nucleotide identity (see "Methods"), resulting in 19,607 clusters. Additional automatic quality control available in PhyloPhlAn 3.0, removed

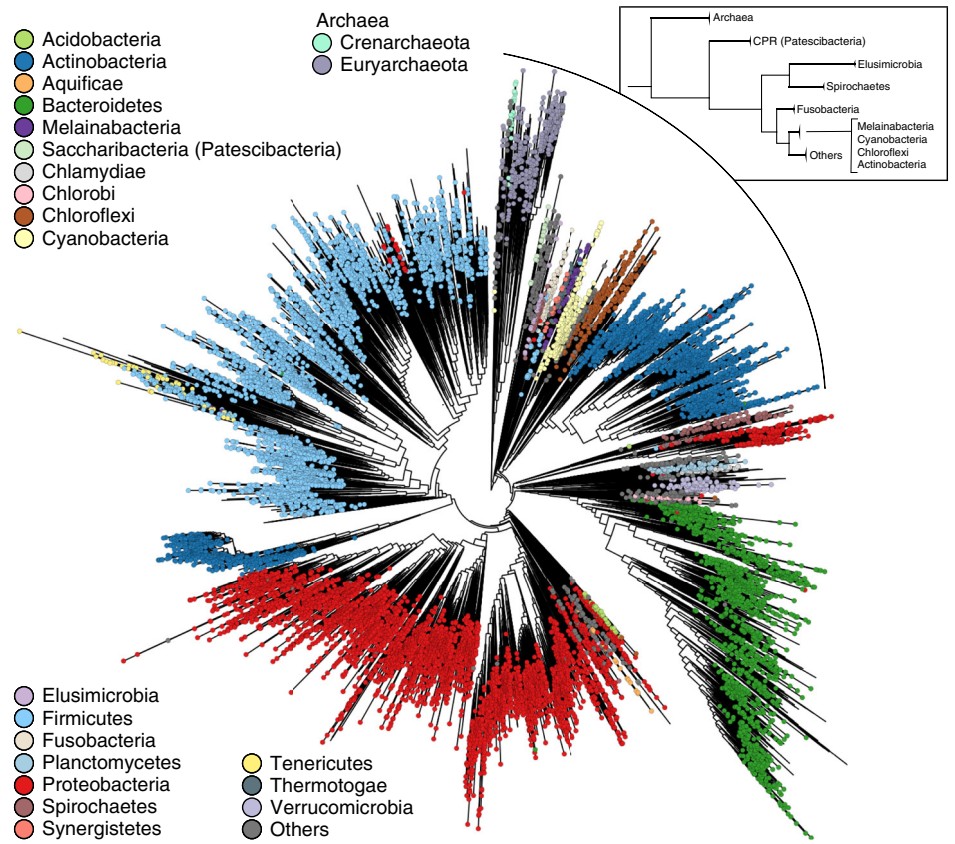

**Fig. 4 PhyloPhlAn 3.0 microbial tree-of-life with 17,672 species-representative genomes from 51 known and 84 candidate phyla.** With 17,672 species-dereplicated isolate genomes and MAGs as input (see "Methods"), PhyloPhlAn 3.0 used 400 optimized universal marker sequences to produce a pan-microbial phylogeny in approximately 10 days (~24,000 CPU-hours on 100 parallel cores). The underlying multiple-sequence alignment comprised 4522 amino acid positions from among 1,872,710 in the untrimmed concatenated marker alignments.

genomes containing less than 100 of PhyloPhlAn's 400 optimized deep-branching marker genes ("Methods"), resulting in 17,672 representative genomes in the final tree. While Proteobacteria are prevalently found in non-human samples, Actinobacteria are instead mainly associated with human samples. Firmicutes and Bacteroidetes are more equally derived from both human and non-human samples, with some preferences in specific subtrees of the two phyla (Supplementary Fig. 5). Reconstruction of this tree of life required ~24,000 CPU-hours (about ten wall-clock days using 100 cores in parallel), of which more than half were needed by IQ-TREE[22] for phylogenetic inference.

The concatenated MSA contained 4522 amino acids out of 1.87 M of total length of the untrimmed concatenated marker sequence alignments. The selection of these most phylogenetically informative positions in the MSA is performed by PhyloPhlAn 3.0 in this aggressive setting for scalability purposes and was validated as we reported elsewhere[26] using the *trident* scoring function[53]. Although phylogenies spanning all the known bacterial and archaeal phyla using more sites and more extensive computation could be used as a default ref.[26], the automatic PhyloPhlAn 3.0 pipeline provides a convenient way to incorporate new MAGs and update genome sets. This is achieved while maintaining high phylogenetic accuracy, as shown by previous clade-specific analyses focusing on organisms from the human microbiome[41], by the overall consistency of the PhyloPhlAn tree with the current reference prokaryotic tree-of-life[26] (Supplementary Fig. 6), and by the comparison of the PhyloPhlAn 3.0 approach of using hundreds of universal markers against other prokaryotic tree-of-life phylogenies based on taxonomy or neighbor-joining (Robinson–Foulds distance < 0.3) reported elsewhere[26]. PhyloPhlAn 3.0 is thus able to efficiently reconstruct extremely large-scale phylogenies, automatically incorporating new isolate genomes, new MAGs, and existing isolate and MAG sequences.

## Discussion

Modern microbial phylogenetic reconstruction methods must, ideally, be extremely versatile: individual sequencing projects can span thousands of closely related isolate genomes, metagenomes can produce hundreds of high-quality but quite diverse MAGs, and cheap sequencing regularly yields thousands of new genomes[41,52,54,55]. Jointly accommodating closely related and deep-branching microbial genomes is thus difficult, especially at large scale. In little more than a decade, pan-microbial phylogenies have grown from 191 genomes[56] to more than 10,000 species[26]. PhyloPhlAn 3.0 provides a combination of methods, precomputed optimized sequences, and study-tunable parameters that accommodate this scale and diversity while also achieving high accuracy for small studies of a few closely related strains.

Here, we showed that PhyloPhlAn 3.0 is accurate across this wide scale of phylogenetic analyses and applications. It allows to construct strain-level phylogenies that automatically include as many reference genomes as needed from public databases, immediately contextualizing newly sequenced isolates. It can further assign a putative taxonomic label based on this phylogenetic placement, both for isolates and MAGs. Comparison of automatically obtained phylogenies with respect to manually curated and evaluated phylogenetic trees showed that PhyloPhlAn 3.0 is highly accurate at different resolutions, ranging from species-level clades to the whole prokaryotic tree-of-life. While for several tasks the fully automatic pipeline should already provide the answer for the problem at hand, our pipeline permits extensive customization of each step for more in depth and personalized analyses. Therefore, we anticipate that PhyloPhlAn 3.0 will serve as a useful instrument to understand present and future microbial diversity in a wide range of microbiological and ecological settings.

## Methods

**Overview of the PhyloPhlAn 3.0 approach**. The PhyloPhlAn 3.0 framework was developed to phylogenetically characterize combinations of isolate genomes, proteomes, and newly reconstructed MAGs. The framework scales to many thousand input sequences and can automatically reconstruct phylogenies at multiple levels of resolution from strain-resolved species-level trees to the scale of the whole microbial tree-of-life. PhyloPhlAn 3.0 integrates public databases by automatically retrieving reference genomes and species-specific sets of UniRef90 proteins. By incorporating over 150,000 MAGs and 80,000 reference genomes that are recapitulated in 17,672 taxonomically labeled SGBs, PhyloPhlAn 3.0 can also assign new input MAGs to species and taxonomic units and phylogenetically refine the corresponding species-level trees. We first describe below the overall phylogenetic pipeline and then detail specific PhyloPhlAn 3.0 approaches to use available reference genomes, retrieve the most appropriate phylogenetic markers, perform taxonomic assignment and refinement, adopt specific choices for very large scale phylogenies, and provide additional information obtained from the resulting phylogenies. Most of these features are unique to PhyloPhlAn 3.0 and were not available in the first version of the framework as detailed in the comparison table Supplementary Data 1.

**The underlying phylogenetic inference pipeline**. PhyloPhlAn 3.0 implements a modular, parallel, and customizable phylogenetic pipeline starting with the detection of phylogenetic markers from the input sequences to the final tree inference. PhyloPhlAn 3.0 modularity allows to parallelize internally to the framework the steps that are independent and can be executed in parallel. Otherwise, PhyloPhlAn 3.0 provides the available number of cores specified by the user to the single program that can then internally exploit the multiprocessing computation. The general pipeline can be divided into four main steps: (i) marker gene identification, (ii) MSA and refinement, (iii) concatenation of MSAs or gene tree inference, and (iv) phylogeny reconstruction.

The marker gene identification step (i) aims at first selecting the most relevant and the highest number of phylogenetic markers for the input sequences and then identifying them in the input sequences. The selection of the markers depends on the type of phylogeny considered and ranges from the 400 universal proteins to a variable number of core genes and species-specific genes (see below). The identification step requires mapping the selected set of markers against the input sequences to extract their homologs. Since both markers and inputs can be a mix of genes (genomes) and proteins (proteomes), this step requires a tool that can optionally perform a translated search. PhyloPhlAn 3.0 currently supports the BLAST suite[27], USEARCH[28], and Diamond[29]. Depending on the type of markers, PhyloPhlAn 3.0 will continue phylogenetic analysis using nucleotide–nucleotide alignment if both markers and inputs are nucleotides, but will proceed with protein or translated mapping if markers are proteins and inputs a mix of genomes and proteomes. The result of this step in PhyloPhlAn 3.0 is the set of marker genes (or proteins) containing the unaligned matching sequences found in the inputs.

Once the markers are identified in the inputs, in step (ii) each variant of each marker is aligned using one of the MSA software available. In PhyloPhlAn 3.0 we included and tested the following tools: MUSCLE[10], MAFFT[11], Opal[13], UPP[30], and PASTA[14]. These tools can differ by performance and accuracy also depending on the configuration parameters of each tool, and while PhyloPhlAn 3.0 adopts MAFFT as default, the user can specify the preferred MSA tool to use in the configuration file. PhyloPhlAn 3.0 is not limited to the software listed above, because other MSA tools can be specified as needed using the configuration file. Moreover, PhyloPhlAn 3.0 includes a set of strategies for quality control and for shortening the alignments, which are discussed in a separate section below. The final results from this step are the MSAs for each marker.

The step (iii) in the overall PhyloPhlAn 3.0 pipeline performs either the concatenation of the MSAs in a unique MSA or the inference of a phylogeny for each MSA. This depends on the choice between a downstream phylogenetic approach based on core-genome maximum-likelihood strategies[6,16,22] or on gene tree-based phylogenetic analysis[18,21]. For the concatenation pipeline, all the computed MSAs are simply merged without a specific order into one large MSA. For the gene trees pipeline, instead, each single MSA is used to compute one phylogeny and the whole set of phylogenies is provided to the downstream tree reconciliation step.

The final step (iv) of PhyloPhlAn 3.0 is the reconstruction of the phylogeny from the concatenated alignments or from the single-gene phylogenies. For both the concatenation-based pipeline and the step of single-gene tree reconstruction for the gene tree pipeline, PhyloPhlAn 3.0 integrates FastTree[16], RAxML[6], IQ-TREE[22], as well as other similar software that the user can specify via configuration files. It also implements the two-steps reconstruction approach[6,57], by deriving the first phylogeny with any of the available approaches and then refine it in a second step using RAxML (or other equivalent software that can be specified in the configuration file). For the gene tree pipeline, the final step of reconciling single-gene trees into genome trees is performed by PhyloPhlAn 3.0 using ASTRAL[18] or ASTRID[21].

**Integration of publicly available microbial genomes**. PhyloPhlAn 3.0 provides the possibility to integrate sets of already available microbial genomes or MAGs to better contextualize the phylogenetic analysis of the user-provided inputs. This compendium of publicly available and taxonomically labeled genomes is increasing and based on UniRef release of 2018_04 (2019_01 in parenthesis) consists of 647 (748) archaeal species with 828 (985) reference genomes, 16,960 (16,638) bacterial species with 86,192 (99,907) reference genomes, and 14 (124) eukaryotic species relevant for the human microbiome analysis with 153 (412) reference genomes. The list of reference genomes for download is compiled by considering those genomes that have a proteome in UniProt and comprises three types of reference genomes: genomes that are considered as reference, non-reference, and redundant by UniProt[58]. Genomes belonging to the set of reference genomes are selected by UniProt as the most well-annotated representative for the species, while a genome is marked as redundant if it is highly similar to another one in the same species[58]. A convenience script is available in the PhyloPhlAn 3.0 package for the download (*phylophlan_get_reference.py*) which guides the user in the choice and number of reference genomes to download and incorporate in the analysis. These reference genomes are sorted according to their classification in UniProt, where the first genomes are marked as *reference*, followed by genomes marked as *non-redundant*, and then all the other available genomes. In this way, PhyloPhlAn 3.0 ensures it will retrieve the genome(s) marked as *reference* first for each taxonomic entry.

**Selection of phylogenetic markers**. The optimality of the genetic markers used for microbial phylogeny reconstruction depends on the diversity and relatedness of the considered genomes. PhyloPhlAn 3.0 extends the default option to use the 400 gene families that are most prevalent across bacterial and archaeal species (i.e., *universal markers*)[1] and that have been recently further validated for use in large scale phylogenetic analysis[26] with species-specific marker genes for each known or candidate species and with the possibility of using user-defined markers.

Species-specific marker genes are those genes found to be core within all the genomes available for the species. These markers are pre-identified based on the UniRef90 protein clusters defined on UniProtKB proteins[35]. Briefly, all genomes are annotated with the UniRef90 catalog and the prevalence of each UniRef90 entry for each species is computed. The set of core UniRef90 families is then defined for each species by selecting those UniRef90 families present in at least 75% of the proteomes available for the species. PhyloPhlAn 3.0 can retrieve automatically the set of such UniRef90 markers for each species of interest without the need for the run-time execution of the pangenome analysis. This retrieval step will be updated roughly every 6 months to include new UniRef90 protein clusters and species. The property of markers of being core within the species is ensured also after the integration of the input genomes, and thus markers that are not consistently found in the analyzed genomes are discarded from the downstream phylogenetic analysis to avoid biases due to partially divergent gene composition in the inputs.

PhyloPhlAn 3.0 can also consider any set of markers computed by the user with different strategies and provided as a fasta sequence file for either amino acids or nucleotides. These markers can be at a higher or lower resolution as those currently provided by the framework and can be integrated using the database setup script (*phylophlan_setup_database.py*).

**PhyloPhlAn 3.0 databases management**. Several convenience scripts are available in PhyloPhlAn 3.0 to handle the databases at different scales and for different analyses. In particular, the scripts *phylophlan_get_reference.py*, *phylophlan_setup_database.py*, and *phylophlan_metagenomic.py* have been developed to handle different database files that are (i) automatically retrieved when needed and only if not present locally, (ii) stored locally after the download, and (iii) updated when the users specify the *--database_update* parameter. Database files comprise the sets of precomputed species-specific UniRef90 proteins, the list of available genomes from GenBank, and the SGB release.

**PhyloPhlAn 3.0 refinement of MSAs**. MSAs need to be quality controlled to avoid local misalignments and alignment positions dominated by missing nucleotides (gaps). To refine an MSA, a number of methods have been proposed[31,59–62] and a recent comparison work[63] suggests that Noisy and trimAl are the best approaches for automatically reducing an MSA. However, when comparing the execution time, trimAl is faster (seconds compared to hours required by Noisy), so we choose to integrate trimAl as an option to trim gappy regions in PhyloPhlAn 3.0. Other approaches for shortening an MSA are the removal of single gaps, the removal of conserved regions with a limited phylogenetic signal, and the removal of extremely variable positions, probably representing low-conserved or noisy regions that result in frequent homoplasies. In PhyloPhlAn 3.0 it is possible to use a combination of the above approaches that have been newly implemented in the software package.

Another more aggressive approach to refine an MSA is scoring each aligned position and then take only a certain number of high-scoring (i.e., phylogenetically relevant) positions. Several different scoring measures have been proposed for evaluating the quality of positions in an MSA[53,64–67] that are also implemented in PhyloPhlAn 3.0 and can be used to reduce MSAs to only a limited number of phylogenetically relevant positions. The three scoring functions available are: *trident*, *muscle*, and *random* that assign a phylogenetic score to each position in the

MSA and, in combination with a subsample function, retain only a certain number of positions. The *random* function simply assigns a random number to each column of the MSA. The *trident* score, as proposed in ref. [53], is a weighted combination of three different measures: symbol diversity, stereochemical diversity, and gap frequency. Gap frequency counts the number of gaps in each column. Symbol diversity measures the entropy of the column by weighting the frequency of each symbol. Stereochemical diversity is a score based on a substitution matrix. In PhyloPhlAn 3.0 we provide four substitution matrices: MIQS[68], PFASUM60[69], VTML200[65], and VTML240 as implemented in MUSCLE[10], along with scripts for generating custom ones. The *muscle* scoring function re-implements the scoring function available in MUSCLE[10] (using the *-scorefile* param). After having scored each position of each MSA, PhyloPhlAn 3.0 uses one of the implemented subsample functions: *phylophlan*, *onethousand*, *sevenhundred*, *fivehundred*, *threehundred*, *onehundred*, *fifty*, *twentyfive*, *tenpercent*, *twentyfive percent*, and *fiftypercent*, to retain only a certain number of positions. The *phylophlan* subsample function is based on the formula in[1] and is specific for the set of 400 universal markers proposed in the same work.

**Pipeline for taxonomic assignment of genomes and MAGs**. One of the novel additions in PhyloPhlAn 3.0 is the assignment of the closest SGBs, a concept and framework we recently introduced[41], to the set of genome bins from MAGs provided as input. This is achieved by using the *phylophlan_metagenomic.py* script that groups the bins based on their closest assigned SGB (configurable using the *--threshold* param, set to 0.05 by default). As the SGB system is continuously updated, also PhyloPhlAn 3.0 will provide the user the possibility to use the latest SGB release available and this is achieved through the *--database_update* parameter as discussed in the *PhyloPhlAn 3.0 databases management* paragraph. The user can then decide to select subsets of inputs and use the *phylophlan_get_reference.py* script to download the needed reference genomes, and *phylophlan_setup_database.py* script in the case of a kSGB to download the core set of UniRef90. For each input MAG, PhyloPhlAn 3.0 reports by default the closest 10 SGBs sorted by their average Mash distance. For each SGB the output includes additional information including whether the SGB comprises or not a taxonomically labeled reference genome, the level at which a confident taxonomic label could be assigned (i.e., species, genus, family, and phylum), the full taxonomic label, and the average Mash distance to the MAGs and genomes in the SGB.

For the cases in which PhyloPhlAn 3.0 cannot assign an SGB to an input genome, the assignment procedure is repeated at the level of GGBs and FGBs. Similarly to SGBs, GGBs, and FGBs were defined elsewhere[41] via hierarchical average linkage clustering at 15% and 30% genetic distance, respectively. These thresholds were empirical estimated in the same work as those more closely reflecting the genetic span of the known taxonomically defined genera and families. GGBs and FGBs are also taxonomically assigned to known genus and family labels if the clusters comprise one or more reference genomes within the corresponding average genetic distance (15% for GGBs, 30% for FGBs, in the case of taxonomic inconsistencies in reference genomes falling inside the same SGB/GGB/FGB, a majority voting approach is applied to assign the most represented taxonomic label). Using this definition of GGBs and FGBs, PhyloPhlAn 3.0 assigns input genomes missing SGB assignment (i.e., the input genome is at >5% average genetic distance with respect to all SGBs) to the closest GGB and/or FGB that are at an average genetic distance <15% and <30%, respectively. If the average genetic distance of the input genome is >30% to any FGBs, limitations in nucleotide similarity quantification methods would not allow reliable higher-level taxonomic assignment[41]. In these cases, PhyloPhlAn 3.0 reports the phylum label of the set of closest reference genomes (i.e., the set of genomes within 5% genetic distance from the closest) decided via majority voting.

**PhyloPhlAn 3.0 strategy to scale to very large phylogenies**. The main challenge when building very large phylogenies is to limit the length of the MSA that will be provided to the inference phylogeny tool. To reduce the length of an MSA, PhyloPhlAn 3.0 exploits the approaches described above and aggressively refines the MSAs to retain only few but phylogenetically meaningful positions in each MSA. The default settings when building very-large phylogenies (parameters: *--diversity high --fast*) are: (i) the application of trimAl[31] (with *-gappyout* param) for the removal of gappy regions, (ii) the removal of conserved regions by considering all positions that do not vary in more than 95% of the inputs (param *--not_variant_threshold 0.95*), and (iii) the removal of the genomes with more than 65% of gaps (*--fragmentary_threshold 0.65*) from the MSA. All these three parameters are automatically set by the *--diversity high --fast* combination.

**Post-phylogeny data and integration with downstream analysis**. PhyloPhlAn 3.0 also provides a set of additional supporting information and visualization accompanying the produced phylogeny. These include the MSAs used to build the phylogeny and the estimated mutation rates between all pairs of inputs. The output of PhyloPhlAn 3.0 can thus be used to additional downstream analysis including, for example, tools for detection and removal of phylogenetic outliers[70] or to perform bootstrapping analyses. PhyloPhlAn-generated trees can be visualized with GraPhlAn[39] and the output of the taxonomic profiling tasks can be displayed with newly implemented scripts in the package.

**Configuration files**. PhyloPhlAn 3.0 uses configuration files that specify both the type and internal choices of the phylogenetic pipeline that will be executed (concatenation or gene trees) and which external tools and parameters to use. Configuration files contain the information about which external software to use and which parameters settings to adopt for them, whereas configuration of the implemented part of the pipeline comprises a wide set of parameters ranging from the type of input (e.g., nucleotides or amino acids), the phylogenetic approach (e.g., concatenation vs. genes tree), and all the steps not available as external applications (e.g., parameters for trimming MSAs). Default configuration files can be generated by scripts present in the software package for standard analysis or as starting points to more ad-hoc and refined pipelines. Integration of new tools not available in the different steps of the framework can be achieved by manually editing the configuration files and inserting the desired tools/parameters, as long as input and output files are in the same format of currently implemented tools. This procedure is described with a dedicated section (*Integrating new tools in the framework*) in the documentation available in the PhyloPhlAn 3.0 code repository. The PhyloPhlAn 3.0 pipeline relies on both the configuration file and the output log generated during the analysis to track which external tools have been used with their specific set of parameters and the details of the execution, to make the obtained results reproducible.

**_Staphylococcus aureus_ and _Escherichia coli_ analyses**. We used PhyloPhlAn 3.0 to generate the phylogenies of 1000 *S. aureus* reference genomes and 135 *S. aureus* isolates as discussed in the results. To evaluate the phylogeny generated by PhyloPhlAn 3.0 we used the tqDist[71] function available in the R *quartet* package to compare quartet distances between the PhyloPhlAn 3.0 and the manually curated reference phylogeny[72].

We used MetaMLST[47] to type 200 reference genomes and eight MAGs from the Ethiopian cohort described in[46], against the University of Warwick MLST schema for *E. coli*. Phylogroups were assigned according to data from Enterobase[73]. An MLST locus was considered detected if a BLAST[27] search against the database of MLST alleles returned a hit covering at least 90% of the locus length at a percentage of identity of 90% or higher. STs were assigned only if all MLST loci could be detected.

**Reporting summary**. Further information on research design is available in the Nature Research Reporting Summary linked to this article.

## Data availability

Raw metagenomes for the Ethiopian cohort are available under NCBI-SRA BioProject id PRJNA504891 [https://www.ncbi.nlm.nih.gov/bioproject/PRJNA504891] and the 369 MAGs can be downloaded from the software page at http://segatalab.cibio.unitn.it/tools/phylophlan.

## Code availability

PhyloPhlAn 3.0 is released open-source and available in GitHub at https://github.com/biobakery/phylophlan and the version used in this work is archived with https://doi.org/10.5281/zenodo.3727181. Manuals and online tutorials describing the PhyloPhlAn 3.0 framework are available at https://github.com/biobakery/phylophlan/wiki. User support is provided both through the issues tracking system in the GitHub repository (https://github.com/biobakery/phylophlan/issues) and the bioBakery help forum (https://forum.biobakery.org).

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

## Acknowledgements

This project has received funding from the European Research Council (ERC) under the European Union's Horizon 2020 research and innovation program (grant agreement No. 716575) to N.S. The work was also supported by MIUR "Futuro in Ricerca" RBFR13EWWI_001 and by the European Union (H2020- SFS-2018-1 project MASTER-818368 and H2020-SC1-BHC project ONCOBIOME-825410) to N.S. This study was also supported by the National Cancer Institute of the National Institutes of Health (1U01CA230551 to N.S.).

## Author contributions

F.A., C.H., and N.S. conceived the method; F.A. implemented the software and performed the experiments; F.A., A.M.T., F.B., C.M., P.M., Q.Z., M.B., F.C., U.M., J.G.S., Se. M., M.Z., E.K., E.P., R.K. Si. M., C.H., and N.S. analyzed the data and provided the feedback, F.A., C.H., and N.S. wrote the paper.

## Competing interests

The authors declare no competing interests.
