## [Peer Review File · Nature Communications]

Reviewers' comments:

Reviewer #1 (Remarks to the Author):

The authors describe PhyloPhlAn v2, an updated bioinformatics pipeline for inferring trees and assigning taxonomic classifications at varying levels of resolution. The manuscript highlights the functionality of PhyloPhlAn with a number of use cases. I believe many in the research community will find PhyloPhlAn v2 to be a valuable resource. I do have some concerns regarding the current manuscript in terms of the clarity of reported results. The manuscript would also benefit from additional proofreading, and the inclusion of page and line numbers would be appreciated.

Major:

1. I am unclear when GGB's would need to be consulted. Doesn't a genome always have a set of closest SGB's even if this aren't particularly close? Is a genome assigned to a GGB if the genomic distance is >15%? This is a bit unusual as ANI values (and especially the estimates provided by Mash which are known to deviate from BLAST-based ANI values for more distance genomes) are generally not recommended for classifications above the rank of species where average amino acid identity is more commonly used.
2. Can the authors comment on the 2.9% of the 1,520 isolate genomes that had a different species assignment from the original study? Is this the result of using Mash to estimate ANI or some other difference relative to the original study?
3. I had trouble following the analysis of the 13 MAGs assigned to an uncharacterized Chlamydiae SGB. It appears the conclusion is that these MAGs belong to a novel phylum. If this is the case, why were they originally assigned to a Chlamydiae SGB? Given that these MAGs aren't Chlamydiae, shouldn't a broader set of phyla be considered in the phylogenetic analyses to confirm these MAGs belong to a novel phylum? What criteria, if any, was used to support this lineage being a novel phylum and not the most basal Chlamydiae lineage?
4. The section on reconstructing the microbial tree-of-life should be expanded? Does the resulting PhyloPhlAn tree agree with previous published results for the included genomes? Both the human MAG studies and the UBA MAGs have NCBI taxonomy strings. Do the taxonomic assignments made by PhyloPhlAn largely agree with these previous assignments? This would be particularly interesting for the UBA dataset where I imagine a number of MAGs are not assigned to SGB's.

Minor:

1. It would help orient reads to explicitly call out the Archaea, CPR (Patescibacteria), and Bacteria in Figure 4.
2. Why are there only ~83,000 GenBank genomes sourced from NCBI? There are >150,000 genomes in the NCBI Assembly Database.
3. How often will the PhyloPhlAn v2 reference databases (e.g. SBGs, species-specific marker sets) be updated? It appears the same set of 400 universal proteins established in 2013 is still being used. This is likely fine, but species-specific sets need to be updated more frequently and I am unclear if this is done de novo or the marker sets precomputed.
4. Given the availability of the Genome Taxonomy Database (GTDB) as an alternative taxonomic framework, it would be good to specifically indicate that PhyloPhlAn makes classifications based on the NCBI Taxonomy. For reference, the latest GTDB reference trees cover >24,000 species.

Reviewer #2 (Remarks to the Author):

In their manuscript, Asnicar et al. present with PhyloPhlAn2 a comprehensive and automated pipeline to analyze microbial genomes in a phylogenetic context. To quickly summarize the main features, PhyloPhlAn2 allows for a collection of taxa, represented by the input genome sequences, the automated selection of a set of phylogenetic markers. Here, it is possible to either choose from 400 'most universal markers', in case the taxa should be embedded into a broad phylogenetic context, or from more than 18,000 sets of pre-selected UniRef90 gene families, if the focus rests on the resolution of phylogenies on a species level. In addition, the user can choose to complement the initial collection of input taxa with existing data selected from more than 150,000 bacterial genomes assembled from metagenomes and from more than 80,000 reference genomes from the public domain. Homologs to the phylogenetic markers are then identified via the BLAST- or BLAST-like search heuristics. Users can then select from a comprehensive toolbox to align the sequences, post-process the alignments and then compute trees from them. Previously uncharacterized taxa can be taxonomically classified, and can thus be embedded into the currently known bacterial diversity. On the example of two showcase studies, the authors then show how PhyloPhlAn2 can be applied in state-of-the-art bacterial comparative genomics analyses. In summary, PhyloPhlAn2 provides a considerably easy way to set the phylogenetic framework for further downstream comparative genomics studies on bacterial taxa, identified either from metagenomes or via the sequencing of isolated clones.

The establishment of standardized, yet flexible workflows for bioinformatics sequence analyses is considerably common, and comprehensive software packages exist for a diverse spectrum of applications. It is inherent in such workflows that they typically don't advance the state of the art in the research field via the implementation of novel algorithms, and this is also true for PhyloPhlAn2, which ships with a considerably conventional default toolbox. Instead, they make large scale comparative studies feasible even for people that otherwise lack the skills to mine and analyze genome data from several thousand taxa. From this aspect, I consider it the main strength of PhyloPhlAn2 that it makes it straightforward to extend the scope of the phylogenetic analysis of custom genome collections to consider the full diversity of genomes available in the public domain. Once the data has been compiled, standard approaches are used for aligning the sequences, for computing phylogenetic trees and for performing a taxonomic classification of, thus far, anonymous genomes. Overall, I consider PhyloPhlAn2 a tool, which is likely to be useful for the scientific community interested in analyzing bacterial genomes in a standard phylogenetic framework.

While I trust that PhyloPhlAn2 can be a helpful pipeline, I also see a number of issues with the presentation in the manuscript.

Major issues

1. In their introduction, the authors implicitly advertise their pipeline by stating that the sequential manual performance of the individual steps in a phylogenetic analysis "requires substantial expertise in identifying the right targets, parameters, and steps (...)". I would like to make a strong point about the fact that the use of standardized workflows, such as PhyloPhlAn2 does not alleviate this burden. A meaningful phylogenetic analysis demands a thorough understanding of the evolutionary processes that shape a phylogeny, and of the algorithms, models and parameters therein that are used to extract the phylogenetic signal from the sequence data.
2. PhyloPhlAn2 ships with a standard toolbox, however, the authors state that it is considerably easy to integrate other software, e.g. for performing the alignments or for tree reconstruction. I consider it essential that precise information about how this integration can work is provided at least in the supplement.
3. In the introduction, the authors state that different genomic regions can be used to achieve a different resolution in differing clades. From the manuscript, I can only see two levels of

resolution. One on the species level using the UniRef90 gene families, and one very general using the 400 most universal marker genes. Given the initial statement, I was expecting to have marker sets also for different taxonomic levels, e.g. genus, family or phylum. Now it appears that this is left to the user to provide the corresponding custom marker sets. In this context, it would be relevant to specify how custom sets of phylogenetic markers can be integrated into the analysis, and to what formatting conventions they have to adhere to.

4. In the manuscript, it is repeatedly stated that PhyloPhlAn2 facilitates a precise placement of genomes and metagenomes. I find this claim mainly supported by the comparison of trees, or more precisely of the patristic distances in the trees, once computed with a non-automated stepwise procedure, and once computed with PhyloPhlAn2. However, I assume that basically the same analysis steps, probably largely overlapping marker sets, and the same software was used. It is, thus, not particularly surprising that the results seem largely congruent. In this context, I find it a bit unfortunate that the actual data supporting the congruence is a correlation analysis of patristic distances between pairs of taxa. I see two problems here. First, differences in the tree topology—which is the relevant measure when it comes to precision of a taxon placement—are probably only very poorly reflected in such an analysis. Second, figure 2B clearly shows that normalized patristic distances—I did not find any explanation how the normalization was performed—are substantially longer in the PhyloPhlAn2 analysis compared to the sequential analysis.

The same consistency argument is then again used in the context of taxonomic classification. Here the authors state that their classification agrees with that which is found in the original work. I am again wondering if the authors can rule out that the classification in the original work was not done using basically the same methods and the same cut-offs as PhyloPhlAn2. Please note that a follow-up manual curation step, which was probably added in the original work re-classified then only a fraction of the taxa.

5. It is not clear to me whether PhyloPhlAn2 directly interacts with the public databases (cf. Fig1), or whether this information is extracted once from the public databases and is then stored locally. If the direct connection is implemented, then I wonder about the times it takes to access and to process the data each time a PhyloPhlAn2 analysis is performed. If the data is downloaded once, then details about this procedure are also highly relevant, e.g. how easy is it to update the data, how much disk space is required. In essence, I think the authors should provide precise information about the data storage and management concept underlying their pipeline.

6. One of the nice things about PhyloPhlAn2 is that genomes from hitherto uncharacterized taxa can be integrated into the analysis and can be subsequently extended via the use of the uSGB concept. It is, however, unclear to me if and how genomes from a previous run of PhyloPhlAn2 can be propagated to the next run where they then can form uSGBs (cf. major issue #5).

7. While phylogenetic trees are considerably dominant in this manuscript, I do not see a single branch support value in the presented trees. I would be interested in the reasons why the authors, other than the rest of the phylogeny community, do not consider support values relevant enough to either provide them, or to explain their absence.

8. The tree shown in Figure 4 is impressive, but I am not sure what the reader should learn from it other than that PhyloPhlAn2 is capable of computing it. As mentioned before, (i) support values are missing, (ii) the tree is based on only 4,522 out of 1,8 Million aligned positions raising the question about whether or not the data is representative, (iii) the resolution makes any meaningful interpretation impossible, which is probably the most severe of the three points. It seems that Firmicutes just as the Euryarchaeota, the Proteobacteria and probably other phyla are paraphyletic. In essence, there is very little to take home from this tree.

9. Given the amount of data that can be potentially analyzed with PhyloPhlAn2, it appears that the hardware of the user comprises a severe bottleneck. It would be relevant to show how the pipeline has been implemented, what kind of parallelization it supports, and how it can be integrated into job scheduling solutions, such as SLURM or SGE.

10. I am not sure how to interpret the sentence in the conclusion, which states that PhyloPhlAn2 automatically includes as many reference genomes as needed from public databases. This implies that the authors have implemented an objective function that helps to automatically select an optimal number of reference genomes for a given analysis. I am really wondering whether this is

indeed the case. If so, I would be highly curious about the way how the authors achieve this.

Minor issues

1. I am confused about the terminology. Are pairwise phylogenetic distances the same as normalized patristic distances. In Figure 2 and in its caption, I find both terms. This should be clarified.
2. It would be great if the authors could provide some additional information to back up the statement "manually checked highly supported species assignments".
3. Please rephrase the statement "assign putative taxonomy". You can classify entities according to a taxonomy, but you cannot assign a taxonomy to an entity.
4. Please explain what you mean with 'phylogenetic neighbor' in subsection 'Phylogenetic context for taxonomically assigned...'
5. When it comes to comparing phylogenetic trees, the authors should refer to standardized metrics to assess the dissimilarity of phylogenetic trees, e.g. the quartet distances, rather than making statements such as 'PhyloPhlAn2's tree was very consistent relative to this previous manually curated phylogeny (...)'
6. I was a bit confused whether 2,127 core genes (figure caption) or 1,658 core genes, which the main text appears to indicate, were used to reconstruct the phylogeny shown in Figure 2A. This should be clarified.
7. In the caption of figure 4, the authors refer to a sequence alignment comprising 4,522 amino acid loci. The use of 'loci' here should be avoided.
8. I do not really understand the distinction between MAGs and genome sequences obtained from bacterial isolates which is made repeatedly in the text. To my understanding, both represent genome sequences representing individual taxa. Why the would one want to make this difference?
9. In the example analysis, the authors reduce the number of taxa for the phylogenetic tree reconstruction by selecting for each species one representative genome. They then test whether the quality of the selected genomes is sufficient to have them included in the tree reconstruction. This results in the exclusion of about 2,000 taxa. I am wondering whether for these excluded 2,000 taxa there is no other representative available that would meet the quality criteria. In essence, why was the quality screen performed after the 'dereplication' and not beforehand?
10. In the methods, the authors refer to high-scoring alignment positions and consider them as 'phylogenetically relevant'. What does this term mean exactly? And how confident are the authors that the scores indeed are linked to something that I would like to call phylogenetic informativeness of an alignment column? For example, I am unaware that Muscle has implemented anything in this direction.
11. In the manuscript, and here in particular with respect to Figs. 2 and 3, it should be clearly specified which analyses have been performed with PhyloPhlAn2, and which have been made separately. If the results shown in figures 2 and 3 have not been generated directly with PhyloPhlAn2, then this should be explicitly stated. Then the authors should explain in the supplementary material in detail how the input data that is required for these analyses can be extracted from PhyloPhlAn2. If they have all been created with this package, then it would be great to see an example workflow in the supplement.

Reviewer #3 (Remarks to the Author):

The paper by Asnicar et al describes the PhyloPhlAn2 software. While we believe the tool is good quality and does important things, we find that the paper (and software) needs some substantial work.

Major and minor points in the paper:

* Nothing serves to unite the family-focused view and the strain-focused view.

* The software has only been used on human microbiome samples so far, we believe? How well does it extend to non-human data?

* The software seems to package many tools that do somewhat overlapping things, which confuses the message in the paper. It would be good to be really clear and explicit about what software is being used for which functionality.

* What is the point of and value of taxonomy here? The paper is largely about assigning taxonomy to MAGs, and it would be good to have a sentence or two about why this is important.

* MAGs are not reference genomes; the two are often conflated in the paper.

* It is unclear why the Ethiopian data was included as a test data set. Is there any motivation beyond wanting to provide an example with unpublished data?

Software questions:

There is little to no detail provided on the software requirements -- much or all of the below should be addressed somewhere.

- * what are the basic memory and compute requirements?
- * how do you modify parameters, if at all?
- * how is traceability and provenance implemented?
- * is the software open source? under what license? this should be mentioned in the paper.
- * how is the software packaged?
- * what is the approach, if any, to automated software testing?
- * what is the approach, if any, to versioning?
- * what are the scalability aspects of the software? does it parallelize nicely?
- * How big is the database on disk, how do you update the database, etc?

The tutorials on the wiki don't seem to work for us.

The files and pipelines mentioned in the paper do not seem to be available.

Reviewer #1 (Remarks to the Author):

The authors describe PhyloPhlAn v2, an updated bioinformatics pipeline for inferring trees and assigning taxonomic classifications at varying levels of resolution. The manuscript highlights the functionality of PhyloPhlAn with a number of use cases. I believe many in the research community will find PhyloPhlAn v2 to be a valuable resource. I do have some concerns regarding the current manuscript in terms of the clarity of reported results. The manuscript would also benefit from additional proofreading, and the inclusion of page and line numbers would be appreciated.

Major:

1. I am unclear when GGB's would need to be consulted. Doesn't a genome always have a set of closest SGB's even if this aren't particularly close? Is a genome assigned to a GGB if the genomic distance is >15%? This is a bit unusual as ANI values (and especially the estimates provided by Mash which are known to deviate from BLAST-based ANI values for more distance genomes) are generally not recommended for classifications above the rank of species where average amino acid identity is more commonly used.

We thank the reviewer for pointing out this, we indeed should not overinterpret GGB assignments. It is also true that when an SGB assignment is found (i.e. the input genome satisfies the threshold to be included in an SGB) then the GGB assignment is straightforward and derived from the GGB of the assigned SGB. However, for the cases in which an SGB is not assigned, then the possible GGB assignment is very relevant. To account for these possibilities in a general way we decided that, to any given genome, even if it is above the 15% genomic distance threshold, the closest GGB will be reported. However, when the genomic distance is above 15% we will not consider that as an assignment, but it is important to report it to give at least some potential phylogenetic/taxonomic context. As an aside, the 15% threshold for GGB assignment was proposed and validated in (Pasolli et al. 2019) in which we show that this threshold maximizes the correct recapitulation of known genera.

To clarify these aspects, we amended the text as follows:

If the input cannot be assigned to any SGB, then PhyloPhlAn2 will report the set of closest SGBs (and their average genomic distances). If needed, this procedure is repeated for higher taxonomic clades with genus-level genome bins (GGBs, up to 15% genomic distance) and family-level genome bins (FGBs, up to 30% genomic distance), ultimately providing a more comprehensive taxonomic context for the set of input genomes to guide downstream analyses and complement their phylogenetic placement.

2. Can the authors comment on the 2.9% of the 1,520 isolate genomes that had a different species assignment from the original study? Is this the result of using Mash to estimate ANI or some other difference relative to the original study?

We thank the reviewer for the comment and we agree it is interesting to analyze the cases in which the species-level taxonomic assignment as provided by the work from Zou et al. is not in agreement with our SGB assignment. We noticed that we should have better clarified in the text the way we did this comparison, in particular, our SGB classification approach can assign a genome or MAG to a species-level cluster, so for doing this comparison of the 1,520 genomes from the Zou et al. work, we selected the 653 that are assigned at species level, according to their **Supplementary Table 5**. The reported number of 97.1% agreement reflects 19 inconsistencies found among 653 genomes with a species-level assignment from the Zou et al. work.

Of the originally identified 19 inconsistencies, we carefully re-checked each case and noticed the following. Here we also report the table of the inconsistencies in **Supplementary Table 2**, with colors to distinguish between the different cases discussed below.

- In two cases (reported in green in the table) the species assignment is actually correct but was considered incorrect only because of inconsistent differences in internal taxonomic labels, i.e. above the species-level. This is only due to changes in the underlying taxonomic system (from NCBI) and should not actually be regarded as differences in the assignment. We have updated the number in the text.
- There are seven cases (reported in blue in the table) where the assigned taxonomic label based on the SGBs is matching at the genus but not at the species level. This is the case of one *Lactococcus garvieae*, which is reported as *L. petauri* in Zou et al., and of six *Megamonas hypermegale*, that in Zou et al. are assigned to *M. funiformis* (three genomes) and *M. rupellensis* (three genomes). We further investigated this difference by computing the average nucleotide identity (ANI) using fastANI between the genomes from Zou et al., the reference genomes for the *L. garvieae* and *L. petauri* in the first case and of *M. funiformis*, *M. rupellensis*, and *M. hypermegale* in the second case, and all the genomes and MAGs belonging to the assigned SGBs. For the genome that we assigned to *L. garvieae*, the closest reference genome confirmed the taxonomic label we assigned (GCA_001645775 - *L. garvieae*, 98.3942% ANI), but also the *L. petauri* reference genome was really close (GCA_002154895, 98.3787% ANI). For the three genomes assigned by Zou et al. to *M. rupellensis*, the taxonomic label was confirmed for two, while the third has been found to be closer to the reference genome of *M. hypermegale*. For the three genomes assigned by Zou et al. to *M. funiformis*, the taxonomic label was confirmed for two of them, while the third, also in this case, has been found to be closer to the reference genome of *M. hypermegale*. This is highlighting that for two out of six genomes reported by Zou et al. the taxonomic label assigned was wrong and instead reported corrected by PhyloPhlAn2. We corrected the number in the text for these two cases. From the further genomic analysis for these three *Megamonas* species, we noticed that based on the Mash ANI distance the reference genomes of the three species are all below the 5% threshold considered the species boundary in the SGB system. This is why PhyloPhlAn2 reports as output only *M. hypermegale* as a putative taxonomic label for the six genomes from Zou et al. This, however, is highlighting a potential taxonomic assignment inconsistency at the level of the reference genomes, so the differences noted by the two approaches here should probably not be considered.

- There are four cases (reported in orange in the table) in which the taxonomic label assigned by Zou et al. (based on EzBioCloud) are four different *Faecalibacterium* sp., while PhyloPhlAn2 found these four genomes falling in the SGB that is taxonomically labelled as *F. prausnitzii*.
- There are six remaining cases (reported in red in the table) in which the two approaches do not agree, where it appears that PhyloPhlAn2 assigns the seven genomes to previously unknown species (uSGB). The Zou et al. approach, instead, reports only the taxonomic label of the closest species, even though the genomes are from a novel and uncharacterized species.

Amended text now reads as follows:

The taxonomic labels inferred by PhyloPhlAn2 were also very consistent (97.7%, **Supplementary Table 2**) with those assigned at species level in the original work highlighting the accuracy of the automatic algorithm.

3. I had trouble following the analysis of the 13 MAGs assigned to an uncharacterized Chlamydiae SGB. It appears the conclusion is that these MAGs belong to a novel phylum. If this is the case, why were they originally assigned to a Chlamydiae SGB? Given that these MAGs aren't Chlamydiae, shouldn't a broader set of phyla be considered in the phylogenetic analyses to confirm these MAGs belong to a novel phylum? What criteria, if any, was used to support this lineage being a novel phylum and not the most basal Chlamydiae lineage?

We apologize if this analysis was hard to follow. The assignment of these MAGs to the Chlamydiae phylum was based on their average Mash distance against the SGB resource. Being these MAGs assigned to an unknown SGB (uSGB, meaning that there are no reference genomes that fall in this SGB within the 5% Mash ANI distance) their assigned taxonomic label was based solely on the phylum of the closest reference genomes. In this case, the closest reference genome was assigned to the phylum Chlamydiae. After the reviewer's suggestion, we enlarged the phylogenetic analysis to a larger set of phyla, based on the set of closest reference genomes that are within 5% of the closest reference genomes. This is a modification we introduced in the taxonomic assignment process that made the procedure more robust for cases of very divergent strains. This expanded the set of phyla of the closest reference genomes for this specific case to Chlamydiae and Proteobacteria, and the class of Epsilonproteobacteria which is not monophyletic with the other Proteobacteria. We then explored again the tree of life we show in **Figure 4** and extended the analysis to up to one reference genome for the set of phyla found close to both Chlamydiae and Epsilonproteobacteria: *Candidatus* Omnitrophica (107 genomes), Chlamydiae (34 genomes), Lentisphaerae (14 genomes), Planctomycetes (57 genomes), Epsilonproteobacteria (196 genomes), Spirochaetes (132 genomes), and Verrucomicrobia (49 genomes), for a total of 589 reference genomes. This new phylogeny (**Supplementary Fig. 4**) shows that uSGB 19436 falls inside Epsilonproteobacteria phylum, in the middle of the genus *Campylobacter*, with bootstrap support of 100. Both the branch length of the subtree and the Mash ANI distance of the MAGs in uSGB 19436 are still showing that they are far from the panel of reference genomes available.

With regards to the definition of a novel phylum, we acknowledge that such claims cannot be made solely on this phylogenetic context and have therefore restructured the text in order to better reflect this:

Phylogenetic context for taxonomically assigned and unassigned input genomes

Since PhyloPhlAn2 associates new genomes and MAGs with SGBs even when the latter do not contain previously characterized taxa, this can be used to automatically compare new genomes and MAGs with hundreds or thousands of phylogenetically related genome sequences (**Fig. 3**). In the Ethiopian study, we focused on the prevalent human gut colonizer *Escherichia coli* (kSGB ID 10068), and on the most prevalent uSGB (ID 19436, 13 MAGs in total) that was assigned to the Proteobacteria phylum. Eight *E. coli* MAGs were constructed from the Ethiopian metagenomes, for which PhyloPhlAn2 retrieved 200 reference genomes and 3,246 UniRef90 families pre-calculated as core to the species (3,099 of which were retained for phylogenetic reconstruction as they are consistently found in the eight input MAGs, **Fig. 3B**, **Supplementary Fig. 3**). This showed the eight Ethiopian input MAGs to be genetically heterogeneous, falling diversely among four different previously defined *E. coli* phylotypes (see **Methods**) based on PhyloPhlAn2-estimated phylogenetic distances (**Fig. 3B** and **Supplementary Fig. 3**). For half of the strains, the placement was confirmed by the phylogroup associated with the MLST types that could be inferred directly on the genomes (Zolfo et al. 2017), but the phylogenetic placement within the clustered phylotypes provides strong evidence for the assignment of the other four strains as well.

We used PhyloPhlAn2 to place the uncharacterized uSGB 19436 in the context of other reference genomes and MAGs from the human microbiome (Pasolli et al. 2019) and of all the automatically-retrieved species' representative genomes from the set of closest phyla that are: part of Proteobacteria (class Epsilonproteobacteria, non-monophyletic with the Proteobacteria phylum), Spirochaetes, Chlamydiae, Planctomycetes, *Candidatus* Omintrophica, Lentisphaerae, and Verrucomicrobia, identified as being close to the Epsilonproteobacteria from the tree of life (**Fig. 4**). PhyloPhlAn2 placed the expanded uSGB 19436 within several very divergent clades taxonomically assigned to the *Campylobacter* genus (**Fig. 3C** and **Supplementary Fig. 4**). The 812 publically available genomes in 108 SGBs assigned to distinct species of *Campylobacter*, reveal this genus to be extremely wide encompassing substantially more than 30% genetic distance (ANI analysis in **Supplementary Fig. 4**) which is a diversity usually characterizing whole classes or orders (Pasolli et al. 2019). Although uSGB 19436 is rooted inside these divergent clades, its genetic divergence (**Supplementary Fig. 4**) is higher than typical family-level divergence and its phylogenetic distance is comparable to the distance between close phyla (**Fig. 3C**) thus confirming PhyloPhlAn2's designation of a new species and genus. The new MAGs from the Ethiopian dataset also reinforce the observation that this phylogenetically divergent uSGB 19436 is specific of non-Westernized lifestyles as the previously reconstructed MAGs from this uSGB are all from populations with rural lifestyles in Madagascar (Pasolli et al. 2019), Peru (Obregon-Tito et al. 2015),

Tanzania (Rampelli et al. 2015), and Bangladesh (David et al. 2015). This analysis thus highlights how PhyloPhlAn2 can be used to expand the phylogenetic diversity of the human microbiome by the simple integration of MAGs from new cohorts in the already large set of microbial genome references considered by the method.

Figure 3. Phylogenetics of MAGs from 50 rural Ethiopian metagenomes using PhyloPhlAn2. (A) Occurrence of the 20 most prevalent SGBs among 50 previously sequenced Ethiopian gut metagenomes highlights the presence of many previously identified but largely uncharacterized species-level genome bins (uSGBs) and the identification of few additional MAGs (unassigned) that are not recapitulated in any already defined SGB. The presence/absence profiles are clustered using average linkage with Euclidean distances. (B) Multidimensional scaling ordination using the t-SNE algorithm on phylogenetic distances from PhyloPhlAn2's tree of eight Ethiopian *E. coli* MAGs (kSGB 10068) integrated with 200 automatically-selected *E. coli* reference genomes using 3,246 UniRef90 gene families for phylogenetic reconstruction. (C) PhyloPhlAn2 phylogeny of Ethiopian MAGs assigned to uSGB ID 19436 including all reference genomes for the closest phyla (589 in total) according to the tree-of-life in Fig. 4. Phylogeny reconstruction used 400 universal markers selected by PhyloPhlAn2 for deep-branching phylogenies. Portion of the tree collapsed are labeled and numbers in parentheses represent the number of genomes in the collapsed sub-trees. Uncollapsed phylogeny is available in Supplementary Fig. 4.

Supplementary Figure 4: Uncollapsed phylogeny of uSGB 19436 with its closest phyla. Uncollapsed phylogeny of the Ethiopian MAGs assigned to the Proteobacteria phylum (class Epsilonproteobacteria) together with the genomes reconstructed in (Pasolli et al. 2019) assigned to uSGB 19436 and the 589 genomes from the closest phyla: Proteobacteria (Epsilonproteobacteria, non-monophyletic with the Proteobacteria), Spirochaetes, Chlamydiae, Planctomycetes, *Candidatus Omintrophica*, Lentisphaerae, and Verrucomicrobia. In the top-right inset, genomes are compared using the average nucleotide identity (ANI) measure.

4. The section on reconstructing the microbial tree-of-life should be expanded? Does the resulting PhyloPhIAn tree agree with previous published results for the included genomes?

Both the human MAG studies and the UBA MAGs have NCBI taxonomy strings. Do the taxonomic assignments made by PhyloPhlAn largely agree with these previous assignments? This would be particularly interesting for the UBA dataset where I imagine a number of MAGs are not assigned to SGB's.

We agree that the new tree-of-life should be better compared with previous efforts. However, this is actually the main point of the back-to-back paper submitted with this manuscript (Zhu et al. 2019) and that has been now published. In the other paper, we use the same markers to build a tree-of-life which is smaller just because we did not include the SGBs with unknown taxonomic assignments (uSGBs). The placement of the uSGBs cannot be tested (in particular with respect to previous trees-of-life) and we thus refer to the other work for the consistency of the tree. This is in line with the main goal of this paper, which is to present and show the potentialities of the PhyloPhlAn2 approach (including the scalability to the largest tree-of-life built) and not to discuss specific phylogenetic results. In the revised paper we thus expanded the tree-of-life paragraph to discuss the comparison with previous effort by pointing at the relevant information of the co-submitted paper.

The edited text is now:

PhyloPhlAn2 can scale to microbial tree-of-life reconstructions including >17,000 genomes and MAGs

In addition to these small-to-medium examples of phylogenetic reconstruction for individual new genome sets, PhyloPhlAn2 can scale to provide automatic placement of thousands of MAGs within the entire current microbial tree of life (**Fig. 4**). Specifically, we considered all high-quality microbial isolate genomes included in UniProt (UniProt Consortium 2014) (87,173 total), >154,000 MAGs from human-associated microbiomes (Pasolli et al. 2019), and ~8,000 MAGs from primarily non-human environments (Parks et al. 2017). These were dereplicated to one representative per species by hierarchical clustering on genomic distances as estimated by Mash (Ondov et al. 2016) with cluster cutoff at 5% intra-cluster nucleotide identity (see **Methods**), resulting in 19,607 clusters. Additional automatic quality control available in PhyloPhlAn2, removed genomes containing less than 100 of PhyloPhlAn2's 400 optimized deep-branching marker genes (**Methods**), resulting in 17,672 representative genomes in the final tree. While Proteobacteria are prevalently found in non-human samples, Actinobacteria are instead mainly associated with human samples. Firmicutes and Bacteroidetes are more equally derived from both human and non-human samples, with some preferences in specific sub-trees of the two phyla (**Supplementary Fig. 5**). Reconstruction of this tree of life required ~24,000 CPU-hours (about ten wall-clock days using 100 cores in parallel), of which more than half were needed by IQ-TREE (Nguyen et al. 2015) for phylogenetic inference.

The concatenated MSA contained 4,522 amino acids out of 1.87M of total length of the untrimmed concatenated marker sequence alignments. The selection of these most phylogenetically informative positions in the MSA is performed by PhyloPhlAn2 in this aggressive setting for scalability purposes and was validated as we reported

elsewhere (Zhu et al. 2019) using the “trident” scoring function (Valdar 2002). Although tree-of-life phylogenies using more sites and more extensive computation could be used as a default reference (Zhu et al. 2019), the automatic PhyloPhlAn2 pipeline provides a convenient way to incorporate new MAGs and update genome sets. This is achieved while maintaining high phylogenetic accuracy, as shown by previous clade-specific analyses focusing on organisms from the human microbiome (Pasolli et al. 2019), by the overall consistency of the PhyloPhlAn2 tree with the current reference tree-of-life (Zhu et al. 2019) (**Supplementary Fig. 6**), and by the comparison of the PhyloPhlAn2 approach of using hundreds of universal markers against other tree-of-life phylogenies based on taxonomy or neighbour-joining (Robinson-Foulds distance <0.3) reported elsewhere (Zhu et al. 2019). PhyloPhlAn2 is thus able to efficiently reconstruct extremely large-scale phylogenies, automatically incorporating new isolate genomes, new MAGs, and existing isolate and MAG sequences.

Minor:

1. It would help orient reads to explicitly call out the Archaea, CPR (Patescibacteria), and Bacteria in Figure 4.

We have added a specific call out for both Archaea and CPR in the new version of **Figure 4**, reported below:

Figure 4. PhyloPhlAn2 efficiently scales to provide a microbial tree of life including 17,672 species-representative genomes spanning 51 known phyla and 84 additional candidate phyla. With 17,672 species-dereplicated isolate genomes and MAGs as input (see **Methods**), PhyloPhlAn2 used 400 optimized universal marker sequences to produce a pan-microbial phylogeny in approximately 10 days (~24,000 CPU-hours on 100 parallel cores). The underlying multiple sequence alignment comprised 4,522 amino acid positions from among 1,872,710 in the untrimmed concatenated marker alignments.

2. Why are there only ~83,000 GenBank genomes sourced from NCBI? There are >150,000 genomes in the NCBI Assembly Database.

We thank the reviewer for pointing out this aspect, which was not clear from the text. Briefly, it is true that there are >150k genomes deposited in NCBI, but in our resource we are considering those genomes for which a proteome is available, as defined in the UniRef resource. This provides a quality-control on the genomes which is crucial for our task. Also, because UniRef performs redundancy reduction on species with thousands of genomes available, this also reduces the number of genomes for over-sequenced species which is not impacting on our SGB-based phylogenies. We thus confirm that using the ~87k genomes from the UniRef release of April 2018 does not represent a real loss of information compared

to the compendium of genomes in the NCBI Assembly Database. Moreover, we have updated the text with the numbers of the new release that is also available in PhyloPhlAn2, which increases the number of reference genomes available for Archaea, Bacteria, and Eukaryotic kingdoms considered.

To make this clearer, with edited the text as follows:

Integration of publicly available microbial genomes

PhyloPhlAn2 provides the possibility to integrate sets of already available microbial genomes or MAGs to better contextualize the phylogenetic analysis of the user-provided inputs. This compendium of publicly available and taxonomically labelled genomes is increasing and based on UniRef release of 2018_04 (2019_01 in parenthesis) consists of 647 (748) archaeal species with 828 (985) reference genomes, 16,960 (16,638) bacterial species with 86,192 (99,907) reference genomes, and 14 (124) eukaryotic species relevant for the human microbiome analysis with 153 (412) reference genomes. The list of reference genomes for download is compiled by considering those genomes that have a proteome in UniProt and comprises three types of reference genomes: genomes that are considered as reference, non-reference, and redundant by UniProt (Bursteinas et al. 2016). Genomes belonging to the set of reference genomes are selected by UniProt as the most well-annotated representative for the species, while a genome is marked as redundant if it is highly similar to another one in the same species (Bursteinas et al. 2016). A convenience script is available in the PhyloPhlAn2 package for the download (`phylophlan_get_reference.py`) which guides the user in the choice and number of reference genomes to download and incorporate in the analysis. These reference genomes are sorted according to their classification in UniProt, where the first genomes are marked as “reference”, followed by genomes marked as “non-redundant”, and then all the other available genomes. In this way, PhyloPhlAn2 ensures it will retrieve the genome(s) marked as “reference” first for each taxonomic entry.

3. How often will the PhyloPhlAn v2 reference databases (e.g. SBGs, species-specific marker sets) be updated? It appears the same set of 400 universal proteins established in 2013 is still being used. This is likely fine, but species-specific sets need to be updated more frequently and I am unclear if this is done de novo or the marker sets precomputed.

The PhyloPhlAn2 species-specific database will be updated roughly every 6 months, following updates to the SGB database (which was just updated in January 2019) and the re-computation of core genes from pangenomes using newly available UniRef90 protein clusters and species. Although the set of 400 universal proteins from 2013 remain the same, their use for large-scale phylogenetic analysis has been recently validated in the companion paper by (Zhu et al. 2019).

This information has been added to the renamed section in the Methods “Selection of phylogenetic markers”:

Selection of phylogenetic markers

The optimality of the genetic markers used for microbial phylogeny reconstruction depends on the diversity and relatedness of the considered genomes. PhyloPhlAn2 extends the default option to use the 400 gene families that are most prevalent across bacterial and archaeal species (i.e. “universal markers”) (Segata et al. 2013) and that have been recently further validated for use in large scale phylogenetic analysis (Zhu et al. 2019) with species-specific marker genes for each known or candidate species and with the possibility of using user-defined markers.

Species-specific marker genes are those genes found to be core within all the genomes available for the species. These markers are pre-identified based on the UniRef90 protein clusters defined on UniProtKB proteins (Suzek et al. 2007). Briefly, all genomes are annotated with the UniRef90 catalog and the prevalence of each UniRef90 entry for each species is computed. The set of core UniRef90 families is then defined for each species by selecting those UniRef90 families present in at least 75% of the proteomes available for the species. PhyloPhlAn2 can retrieve automatically the set of such UniRef90 markers for each species of interest without the need for the run-time execution of the pangenome analysis. This retrieval step will be updated roughly every 6 months to include new UniRef90 protein clusters and species. The property of markers of being core within the species is ensured also after the integration of the input genomes, and thus markers that are not consistently found in the analyzed genomes are discarded from the downstream phylogenetic analysis to avoid biases due to partially divergent gene composition in the inputs.

PhyloPhlAn2 can also consider any set of markers computed by the user with different strategies and provided as a fasta sequence file for either amino acids or nucleotides. These markers can be at a higher or lower resolution as those currently provided by the framework and can be integrated using the database setup script (`phylophlan2_setup_database.py`).

4. Given the availability of the Genome Taxonomy Database (GTDB) as an alternative taxonomic framework, it would be good to specifically indicate that PhyloPhlAn makes classifications based on the NCBI Taxonomy. For reference, the latest GTDB reference trees cover >24,000 species.

We agree with the reviewer and have indicated in the introduction that PhyloPhlAn2 performs classifications based on NCBI taxonomy. The text now reads:

Compared to available alternatives such as the Genome Taxonomy Database (GTDB) (Parks et al. 2018), PhyloPhlAn2 is the only approach able to automatically perform taxonomic assignment of MAGs based on the NCBI taxonomy and to consider unnamed and uncharacterized species in the genomic contextualization task.

Reviewer #2 (Remarks to the Author):

In their manuscript, Asnicar et al. present with PhyloPhlAn2 a comprehensive and automated pipeline to analyze microbial genomes in a phylogenetic context. To quickly summarize the main features, PhyloPhlAn2 allows for a collection of taxa, represented by the input genome sequences, the automated selection of a set of phylogenetic markers. Here, it is possible to either choose from 400 'most universal markers', in case the taxa should be embedded into a broad phylogenetic context, or from more than 18,000 sets of pre-selected UniRef90 gene families, if the focus rests on the resolution of phylogenies on a species level. In addition, the user can choose to complement the initial collection of input taxa with existing data selected from more than 150,000 bacterial genomes assembled from metagenomes and from more than 80,000 reference genomes from the public domain. Homologs to the phylogenetic markers are then identified via the BLAST- or BLAST-like search heuristics. Users can then select from a comprehensive toolbox to align the sequences, post-process the alignments and then compute trees from them. Previously uncharacterized taxa can be taxonomically classified, and can thus be embedded into the currently known bacterial diversity. On the example of two showcase studies, the authors then show how PhyloPhlAn2 can be applied in state-of-the-art bacterial comparative genomics analyses. In summary, PhyloPhlAn2 provides a considerably easy way to set the phylogenetic framework for further downstream comparative genomics studies on bacterial taxa, identified either from metagenomes or via the sequencing of isolated clones.

The establishment of standardized, yet flexible workflows for bioinformatics sequence analyses is considerably common, and comprehensive software packages exist for a diverse spectrum of applications. It is inherent in such workflows that they typically don't advance the state of the art in the research field via the implementation of novel algorithms, and this is also true for PhyloPhlAn2, which ships with a considerably conventional default toolbox. Instead, they make large scale comparative studies feasible even for people that otherwise lack the skills to mine and analyze genome data from several thousand taxa. From this aspect, I consider it the main strength of PhyloPhlAn2 that it makes it straightforward to extend the scope of the phylogenetic analysis of custom genome collections to consider the full diversity of genomes available in the public domain. Once the data has been compiled, standard approaches are used for aligning the sequences, for computing phylogenetic trees and for performing a taxonomic classification of, thus far, anonymous genomes. Overall, I consider PhyloPhlAn2 a tool, which is likely to be useful for the scientific community interested in analyzing bacterial genomes in a standard phylogenetic framework.

While I trust that PhyloPhlAn2 can be a helpful pipeline, I also see a number of issues with the presentation in the manuscript.

Major issues

1. In their introduction, the authors implicitly advertise their pipeline by stating that the sequential manual performance of the individual steps in a phylogenetic analysis "requires substantial expertise in identifying the right targets, parameters, and steps (...)". I would like to make a strong point about the fact that the use of standardized workflows, such as

PhyloPhlAn2 does not alleviate this burden. A meaningful phylogenetic analysis demands a thorough understanding of the evolutionary processes that shape a phylogeny, and of the algorithms, models and parameters therein that are used to extract the phylogenetic signal from the sequence data.

We definitely agree that an accurate and meaningful phylogenetic analysis requires a good understanding of all the steps. However, it is also true that phylogenetics is becoming increasingly available to researchers in microbiology not very familiar with each step; when using it as a tool for providing phylogenetic context to completely unknown genomes (e.g. from uncharacterized microbial isolates or from MAGs), it is arguably true that a phylogeny obtained with an automatic pipeline (with appropriate checks) is an important analysis step that allows non-expert users to obtain phylogenetic information that would not be obtained in other ways. There are several users that are now using the old PhyloPhlAn version to inform phylogenetic/taxonomic analysis and for which the alternative would be to perform a best-match assignment based on the 16S rRNA gene of the genome or a rough BLAST of the whole genome. While we acknowledge that more understanding of the phylogenetic analysis should be acquired by users, we also think that PhyloPhlAn2 with an automatic workflow and few rather general parameters can be very useful especially for the non-very-expert users.

PhyloPhlAn2 does permit full control on each step to the expert user, but again it also provides an automatic pipeline for those non-expert users that would be unable to manually tune each step separately but will nonetheless benefit for the automatically produced phylogenies. Additionally, PhyloPhlAn2 also provide a framework that automatically takes care of input/output formats required by the different tools and a unified access to the naming conventions of the parameters of each tool. We incorporate the reviewer point and suggestion in multiple points of the manuscript in which we highlight the limitation of fully automatic phylogenetic pipelines:

Each tool can be separately and sequentially applied providing full step-by-step control on the whole phylogenetic analysis, but doing so requires substantial expertise not only in identifying the right targets, parameters, and steps for computational phylogenetics, but also in understanding how such tools should be interfaced one with the other.

Comparison of automatically obtained phylogenies with respect to manually curated and evaluated phylogenetic trees showed that PhyloPhlAn2 is highly accurate at different resolutions, ranging from species-level clades to the whole tree-of-life. While for several tasks the fully automatic pipeline should already provide the answer for the problem at hand, our pipeline permits extensive customization of each step for more *in-depth* and personalized analyses. Therefore, we anticipate that PhyloPhlAn2 will serve as a useful instrument to understand present and future microbial diversity in a wide range of microbiological and ecological settings.

2. PhyloPhlAn2 ships with a standard toolbox, however, the authors state that it is considerably easy to integrate other software, e.g. for performing the alignments or for tree reconstruction. I consider it essential that precise information about how this integration can work is provided at least in the supplement.

We agree with the reviewer that a more detailed explanation about how PhyloPhlAn2 integrates additional software is needed. In the new version of the manuscript, we have explained how this integration is performed, that is, through editing or writing a configuration file to include the use of different tools to perform the alignments and/or tree reconstruction steps.

This information was added to the “Configuration files” section of the methods, which reads:

Integration of new tools not available in the different steps of the framework can be achieved by manually editing the configuration files and inserting the desired tools/parameters, as long as input and output files are in the same format of currently implemented tools. This procedure is described with a dedicated section (“Integrating new tools in the framework”) in the documentation available in the PhyloPhlAn2 code repository.

We have also added a section in the Wiki documentation providing an example of how to incorporate the use of Clustal Omega, a multiple sequence aligner not originally used in the framework, which can be seen here:

<https://bitbucket.org/nsegata/phylophlan/wiki/phylophlan2#markdown-header-integrating-new-tools-in-the-framework>

3. In the introduction, the authors state that different genomic regions can be used to achieve a different resolution in differing clades. From the manuscript, I can only see two levels of resolution. One on the species level using the UniRef90 gene families, and one very general using the 400 most universal marker genes. Given the initial statement, I was expecting to have marker sets also for different taxonomic levels, e.g. genus, family or phylum. Now it appears that this is left to the user to provide the corresponding custom marker sets. In this context, it would be relevant to specify how custom sets of phylogenetic markers can be integrated into the analysis, and to what formatting conventions they have to adhere to.

The reviewer is correct, PhyloPhlAn2 currently provides a set of either 400 universal markers or species-specific markers. However, the framework also allows users to provide their own set of phylogenetic markers and create their own database using a custom script (phylophlan2_setup_database.py). The reference to the use of different genomic regions in the introduction refers to the fact that when using the set of 400 universal markers, users can opt to include more positions in the MSA in order to increase resolution.

In order to make these points clear to readers, we have renamed the methods section “Selection of species-specific phylogenetic markers” to “Selection of phylogenetic markers”

and divided it into 3 subsections where each type of phylogenetic marker is discussed; the universal, species-specific, and user-provided.

We transcribe the modified text below:

Selection of phylogenetic markers

The optimality of the genetic markers used for microbial phylogeny reconstruction depends on the diversity and relatedness of the considered genomes. PhyloPhlAn2 extends the default option to use the 400 gene families that are most prevalent across bacterial and archaeal species (i.e. “universal markers”) (Segata et al. 2013) and that have been recently further validated for use in large scale phylogenetic analysis (Zhu et al. 2019) with species-specific marker genes for each known or candidate species and with the possibility of using user-defined markers.

Species-specific marker genes are those genes found to be core within all the genomes available for the species. These markers are pre-identified based on the UniRef90 protein clusters defined on UniProtKB proteins (Suzek et al. 2007). Briefly, all genomes are annotated with the UniRef90 catalog and the prevalence of each UniRef90 entry for each species is computed. The set of core UniRef90 families is then defined for each species by selecting those UniRef90 families present in at least 75% of the proteomes available for the species. PhyloPhlAn2 can retrieve automatically the set of such UniRef90 markers for each species of interest without the need for the run-time execution of the pangenome analysis. This retrieval step will be updated roughly every 6 months to include new UniRef90 protein clusters and species. The property of markers of being core within the species is ensured also after the integration of the input genomes, and thus markers that are not consistently found in the analyzed genomes are discarded from the downstream phylogenetic analysis to avoid biases due to partially divergent gene composition in the inputs.

PhyloPhlAn2 can also consider any set of markers computed by the user with different strategies and provided as a fasta sequence file for either amino acids or nucleotides. These markers can be at a higher or lower resolution as those currently provided by the framework and can be integrated using the database setup script (`phylophlan2_setup_database.py`).

4. In the manuscript, it is repeatedly stated that PhyloPhlAn2 facilitates a precise placement of genomes and metagenomes. I find this claim mainly supported by the comparison of trees, or more precisely of the patristic distances in the trees, once computed with a non-automated stepwise procedure, and once computed with PhyloPhlAn2. However, I assume that basically the same analysis steps, probably largely overlapping marker sets, and the same software was used. It is, thus, not particularly surprising that the results seem largely congruent. In this context, I find it a bit unfortunate that the actual data supporting the congruence is a correlation analysis of patristic distances between pairs of taxa. I see two problems here. First, differences in the tree topology—which is the relevant measure when it

comes to precision of a taxon placement—are probably only very poorly reflected in such an analysis. Second, figure 2B clearly shows that normalized patristic distances—I did not find any explanation how the normalization was performed—are substantially longer in the PhyloPhlAn2 analysis compared to the sequential analysis.

The same consistency argument is then again used in the context of taxonomic classification. Here the authors state that their classification agrees with that which is found in the original work. I am again wondering if the authors can rule out that the classification in the original work was not done using basically the same methods and the same cut-offs as PhyloPhlAn2. Please note that a follow-up manual curation step, which was probably added in the original work re-classified then only a fraction of the taxa.

We agree with the reviewer, that the comparison of the patristic distances is not the best way to perform the comparative evaluation and that tree topology is an important aspect to include in the comparison between the two phylogenetic methods. We thus assessed differences in tree topology and found that there was a 90.5% consistency based on quartet score. We have added this information to the text and added a new panel in **Supplementary Figure 2 (Supplementary Fig. 2C)**:

When comparing PhyloPhlAn2's tree with the previous manually curated phylogeny, we found an overall correlation of 0.992 (Pearson's correlation) between normalized pairwise branch length distances from the two phylogenies (**Fig. 2B**) and 90.5% consistency between quartet distances (**Supplementary Fig. 2**).

Pairwise phylogenetic distances in **Fig. 2B** were normalized by the total branch length of the two trees, respectively. This means that the slight difference in the range of the axis is reflecting only a 5% difference in the branch length by using PhyloPhlAn2 and only affects the largest distances. This can be due to the different sets of core proteins used. PhyloPhlAn2 indeed relies on precomputed sets of UniRef90 proteins, while the manual approach identified the ad hoc set of core genes from the input genomes using the Roary pipeline.

Supplementary Figure 2. Side-by-side comparison of *Staphylococcus aureus* phylogenies. (A) The original phylogeny manually curated (Manara et al. 2018). (B) The phylogeny automatically reconstructed by PhyloPhlAn2 and visualized with GraPhlAn using the same annotations of (A). (C) Ternary plot of quartet distances showing the consistency between *Staphylococcus aureus* phylogenies.

5. It is not clear to me whether PhyloPhlAn2 directly interacts with the public databases (cf. Fig1), or whether this information is extracted once from the public databases and is then stored locally. If the direct connection is implemented, then I wonder about the times it takes to access and to process the data each time a PhyloPhlAn2 analysis is performed. If the data is downloaded once, then details about this procedure are also highly relevant, e.g. how easy is it to update the data, how much disk space is required. In essence, I think the

authors should provide precise information about the data storage and management concept underlying their pipeline.

We thank the reviewer for pointing out this and we have now clarified it in the text. In summary, PhyloPhlAn2 uses a hybrid approach, meaning the needed files are retrieved automatically from public databases and after that, they are stored locally. So, for example, when the user wants to download all reference genomes of *Escherichia coli* and if they were not already downloaded in previous runs, PhyloPhlAn2 will first retrieve a table associating the taxonomic labels to the set of genomes that will then use to automatically download from GenBank the genomes of the species of interest. The table is stored locally until explicitly specified by the user by using the "--database_update" parameter to update the local copy of the file.

The running time for downloading the 200 *E. coli* reference genomes used in **Supplementary Fig. 3** and in the tutorial n. 4 of PhyloPhlAn2, took less than 4 minutes, although this might depend on the Internet connection speed. To give another example, the retrieval of the core set of UniRef90 proteins (3,246 in total) and database creation for *E. coli* as used in **Supplementary Fig. 3** and the tutorial n. 4, took less than 25 minutes.

The disk usage is heavily dependent on the amount of data downloaded by the user. To give an idea, the reference genomes together with the set of UniRef90 proteins use about ~100 Mb, while the SGB database used by phylophlan2_metagenomic.py will be around 10 Gb of disk space. The management of the databases stored locally by PhyloPhlAn2 is done through the parameter "--database_update", implemented in the phylophlan2_get_reference.py, phylophlan2_setup_database.py, and phylophlan2_metagenomic.py scripts, which allows to update the locally stored copy of the files that allow the retrieval of either the reference genomes or the set of core UniRef90 proteins.

The added section to the Methods is reported below:

PhyloPhlAn2 databases management

Several convenience scripts are available in PhyloPhlAn2 to handle the databases at different scales and for different analyses. In particular, the scripts phylophlan2_get_reference.py, phylophlan2_setup_database.py, and phylophlan2_metagenomic.py have been developed to handle different database files that are (i) automatically retrieved when needed and only if not present locally, (ii) stored locally after the download, and (iii) updated when the users specify the "--database_update" parameter. Database files comprise the set of pre-computed species-specific set of UniRef90 proteins, the set of available genomes from GenBank, and the SGB release.

6. One of the nice things about PhyloPhlAn2 is that genomes from hitherto uncharacterized taxa can be integrated into the analysis and can be subsequently extended via the use of the uSGB concept. It is, however, unclear to me if and how genomes from a previous run of

PhyloPhlAn2 can be propagated to the next run where they then can form uSGBs (cf. major issue #5).

PhyloPhlAn2 does not have a direct way to define new uSGBs based on input genomes that are not already part of any SGB. We do have in place a methodology to do so, but if multiple users apply that independently all the downstream analyses can be inconsistent making the new taxonomic assignments not reproducible across datasets. For this reason, we are regularly updating the original SGB resource by Pasolli et al. with all the new genomes and MAGs that we are able to retrieve. The database now consists of more than 300,000 MAGs+genomes with an increase of more than 20%. Regular updates will be available as also discussed in “Pipeline for taxonomic assignment of genomes and MAGs” paragraph in the **Methods**:

As the SGB system is continuously updated, also PhyloPhlAn2 will provide the user the possibility to use the latest SGB release available and this is achieved through the “--database_update” parameter as discussed in the “PhyloPhlAn2 databases management” paragraph.

7. While phylogenetic trees are considerably dominant in this manuscript, I do not see a single branch support value in the presented trees. I would be interested in the reasons why the authors, other than the rest of the phylogeny community, do not consider support values relevant enough to either provide them, or to explain their absence.

We apologize that no support values were added to the trees in the manuscript and re-iterate that their presence is highly relevant for phylogenetic analysis. We would like to point out that although FastTree does provide an estimation of support values for branches, in the current workflow of PhyloPhlAn2 this step is performed before RAxML and these values are not used. However, RAxML can provide support values if bootstrap analysis is performed and this parameter can be added to the configuration file.

In the revised version of the manuscript, we performed bootstrap analysis for both trees in **Fig. 2** and **3**, and added support values for branches with bootstrap >80% in both figures. The computational cost to calculate bootstrap support values for the tree of life presented in **Fig. 4** is unfeasible given the size of the phylogeny and therefore has not been computed.

Figure 2. Accurate reconstruction of *Staphylococcus aureus* phylogenies using PhyloPhlAn2. (A) Phylogenetic tree of 135 *S. aureus* strains from a pediatric hospital (Manara et al. 2018) reconstructed by PhyloPhlAn2 using 2,127 automatically identified core genes (rendered by GraPhlAn (Asnicar et al. 2015) see **Supplementary Fig. 2** for a full comparison). Green circles represent the methicillin-sensitive *S. aureus* (MSSA) while red circles represent methicillin-resistant *S. aureus* (MRSA). Blue circles internal to the phylogeny identify subtrees with bootstrap >80%. (B) Normalized phylogenetic distances in the PhyloPhlAn2-reconstructed tree and in a manually curated phylogeny from (Manara et al. 2018) highlighting strong consistency between the automated PhyloPhlAn2 results and the curated tree (0.992 Pearson correlation coefficient). (C) Multidimensional scaling ordination of pairwise phylogenetic distances from the tree integrating the 135 *S. aureus* isolates (crosses) with 1,000 automatically-selected *S. aureus* reference genomes (circles, **Supplementary Fig. 1**). The ten most prevalent sequence types (STs) (Maiden et al. 1998) are highlighted in different colors.

Figure 3. Phylogenetics of MAGs from 50 rural Ethiopian metagenomes using PhyloPhlAn2. (A) Occurrence of the 20 most prevalent SGBs among 50 previously sequenced Ethiopian gut metagenomes highlights the presence of many previously identified but largely uncharacterized species-level genome bins (uSGBs) and the identification of few additional MAGs (unassigned) that are not recapitulated in any already defined SGB. The presence/absence profiles are clustered using average linkage with Euclidean distances. (B) Multidimensional scaling ordination using the t-SNE algorithm on phylogenetic distances from PhyloPhlAn2's tree of eight Ethiopian *E. coli* MAGs (kSGB 10068) integrated with 200 automatically-selected *E. coli* reference genomes using 3,246 UniRef90 gene families for phylogenetic reconstruction. (C) PhyloPhlAn2 phylogeny of Ethiopian MAGs assigned to uSGB ID 19436 including all reference genomes for the closest phyla (589 in total) according to the tree-of-life in Fig. 4. Phylogeny reconstruction used 400 universal markers selected by PhyloPhlAn2 for deep-branching phylogenies. Portion of the tree collapsed are labeled and numbers in parentheses represent the number of genomes in the collapsed sub-trees. Uncollapsed phylogeny is available in **Supplementary Fig. 4**.

8. The tree shown in Figure 4 is impressive, but I am not sure what the reader should learn from it other than that PhyloPhlAn2 is capable of computing it. As mentioned before, (i) support values are missing, (ii) the tree is based on only 4,522 out of 1,8 Million aligned positions raising the question about whether or not the data is representative, (iii) the resolution makes any meaningful interpretation impossible, which is probably the most severe of the three points. It seems that Firmicutes just as the Euryarchaeota, the Proteobacteria and probably other phyla are paraphyletic. In essence, there is very little to take home from this tree.

We thank the reviewer for their comment and would like to clarify that the purpose of this analysis/figure is to show readers that PhyloPhlAn2 can enable users to create very large phylogenies with little to no manual curation. On the interpretation of the results, we apologize if this was not stressed enough, but this is the main point of the back-to-back paper (Zhu et al. 2019) submitted with this paper. We now refer and discuss more the main point of the other work regarding the interpretation of the three which is also computed without support values for computational reasons and using the same number of positions.

Amended text:

PhyloPhlAn2 can scale to microbial tree-of-life reconstructions including >17,000 genomes and MAGs

In addition to these small-to-medium examples of phylogenetic reconstruction for individual new genome sets, PhyloPhlAn2 can scale to provide automatic placement of thousands of MAGs within the entire current microbial tree of life (**Fig. 4**). Specifically, we considered all high-quality microbial isolate genomes included in UniProt (UniProt Consortium 2014) (87,173 total), >154,000 MAGs from human-associated microbiomes (Pasolli et al. 2019), and ~8,000 MAGs from primarily non-human environments (Parks et al. 2017). These were dereplicated to one representative per species by hierarchical clustering on genomic distances as estimated by Mash (Ondov et al. 2016) with cluster cutoff at 5% intra-cluster nucleotide identity (see **Methods**), resulting in 19,607 clusters. Additional automatic quality control available in PhyloPhlAn2, removed genomes containing less than 100 of PhyloPhlAn2's 400 optimized deep-branching marker genes (**Methods**), resulting in 17,672 representative genomes in the final tree. While Proteobacteria are prevalently found in non-human samples, Actinobacteria are instead mainly associated with human samples. Firmicutes and Bacteroidetes are more equally derived from both human and non-human samples, with some preferences in specific sub-trees of the two phyla (**Supplementary Fig. 5**). Reconstruction of this tree of life required ~24,000 CPU-hours (about ten wall-clock days using 100 cores in parallel), of which more than half were needed by IQ-TREE (Nguyen et al. 2015) for phylogenetic inference.

The concatenated MSA contained 4,522 amino acids out of 1.87M of total length of the untrimmed concatenated marker sequence alignments. The selection of these most phylogenetically informative positions in the MSA is performed by PhyloPhlAn2 in this aggressive setting for scalability purposes and was validated as we reported elsewhere (Zhu et al. 2019) using the "trident" scoring function (Valdar 2002). Although tree-of-life phylogenies using more sites and more extensive computation could be used as a default reference (Zhu et al. 2019), the automatic PhyloPhlAn2 pipeline provides a convenient way to incorporate new MAGs and update genome sets. This is achieved while maintaining high phylogenetic accuracy, as shown by previous clade-specific analyses focusing on organisms from the human microbiome (Pasolli et al. 2019), by the overall consistency of the PhyloPhlAn2 tree with the current reference tree-of-life (Zhu et al. 2019) (**Supplementary Fig. 6**), and by the comparison of the PhyloPhlAn2 approach of using hundreds of universal markers

against other tree-of-life phylogenies based on taxonomy or neighbour-joining (Robinson-Foulds distance <0.3) reported elsewhere (Zhu et al. 2019). PhyloPhlAn2 is thus able to efficiently reconstruct extremely large-scale phylogenies, automatically incorporating new isolate genomes, new MAGs, and existing isolate and MAG sequences.

9. Given the amount of data that can be potentially analyzed with PhyloPhlAn2, it appears that the hardware of the user comprises a severe bottleneck. It would be relevant to show how the pipeline has been implemented, what kind of parallelization it supports, and how it can be integrated into job scheduling solutions, such as SLURM or SGE.

PhyloPhlAn2 has been implemented based on Python 3. It has been designed as a modular pipeline that is able to parallelize the steps that can be carried out independently, like the quality control of the inputs, the orthologous search of the markers, the multiple-sequence alignment of the markers, and the trimming of the aligned markers. For such steps that cannot be carried out independently, we are passing the number of CPUs specified by the user (otherwise the default is 1, so no parallelization) to the external tools that can exploit multi-CPU, like the phylogeny reconstruction and the reconciliation step (in the case of a gene tree pipeline).

The parallelization within PhyloPhlAn2 has been implemented using the multiprocessing package available in Python. PhyloPhlAn2 has not been designed to be run on a system managed by a scheduler. However, the expert user can process offline using a job scheduling system some of the internal steps, and then run PhyloPhlAn2 to perform the remaining parts of the pipeline, exploiting the PhyloPhlAn2 capability of re-starting an analysis by computing only the steps not yet done.

We amend the text to be more clear about this aspect, as follows:

PhyloPhlAn2 modularity allows to parallelize internally to the framework the steps that are independent and can be executed in parallel. Otherwise, PhyloPhlAn2 pass on the available number of cores specified by the user to the single program that can then internally exploit the multi-processing computation.

10. I am not sure how to interpret the sentence in the conclusion, which states that PhyloPhlAn2 automatically includes as many reference genomes as needed from public databases. This implies that the authors have implemented an objective function that helps to automatically selects an optimal number of reference genomes for a given analysis. I am really wondering whether this is indeed the case. If so, I would be highly curious about the way how the authors achieve this.

We thank the reviewer for pointing out this, which was probably not clear and we now have improved the text and the **Methods** section to be more clear about the way genomes are considered when downloading.

PhyloPhlAn2 can retrieve as many genomes as the ones available from GenBank by relying on a mapping file that associates a given taxonomy to a list of reference genomes. The reference genomes are sorted according to their classification in UniProt, where first there are genomes marked as “reference”, followed by genomes marked as “non-redundant”, and then all the other available genomes. In this way, we ensure that even when the user asks to a minimum number of genomes to be downloaded ($n=1$), PhyloPhlAn2 will retrieve first the genome(s) marked as “reference” for each taxonomic entry.

The text has been edited as follows:

A convenience script is available in the PhyloPhlAn2 package for the download (`phylophlan_get_reference.py`) which guides the user in the choice and number of reference genomes to download and incorporate in the analysis. These reference genomes are sorted according to their classification in UniProt, where the first genomes are marked as “reference”, followed by genomes marked as “non-redundant”, and then all the other available genomes. In this way, PhyloPhlAn2 ensures it will retrieve the genome(s) marked as “reference” first for each taxonomic entry.

Minor issues

1. I am confused about the terminology. Are pairwise phylogenetic distances the same as normalized patristic distances. In Figure 2 and in its caption, I find both terms. This should be clarified.

We apologize for the confusion. The two distances are equivalent, patristic distance is defined as the sum of the branch lengths that connect two nodes, which is the same as the phylogenetic distance here. We modified accordingly the two occurrences of patristic distance that appeared in the paper.

2. It would be great if the authors could provide some additional information to back up the statement “manually checked highly supported species assignments”.

We agree with the Reviewer that the sentence was convoluted, and we amended the text as we expanded the analysis while replying to question #2 of Reviewer #1:

The taxonomic labels inferred by PhyloPhlAn2 were also very consistent (97.7%, **Supplementary Table 2**) with those assigned at species level in the original work highlighting the accuracy of the automatic algorithm.

Since with our SGB approach the taxonomic label is assigned at species-level, we tested whether PhyloPhlAn2 was in agreement with the results by Zou *et al.*, in which a different approach to assign taxonomic labels was used (EzBioCloud).

3. Please rephrase the statement “assign putative taxonomy”. You can classify entities according to a taxonomy, but you cannot assign a taxonomy to an entity.

We thank the reviewer and we amended the text as follows:

In addition to phylogenetic reconstruction, PhyloPhlAn2 can assign a putative taxonomic label to new, uncharacterized genomes if they can be confidently placed in well-labeled phylogenetic clades.

Here, we showed that PhyloPhlAn2 is accurate across this wide scale of phylogenetic analyses and applications. It allows to construct strain-level phylogenies that automatically include as many reference genomes as needed from public databases, immediately contextualizing newly-sequenced isolates. It can further assign a putative taxonomic label based on this phylogenetic placement, both for isolates and MAGs.

4. Please explain what you mean with ‘phylogenetic neighbor’ in subsection ‘Phylogenetic context for taxonomically assigned...’

We rephrased “phylogenetic neighbor” with “phylogenetically related genome sequences”:

Since PhyloPhlAn2 associates new genomes and MAGs with SGBs even when the latter do not contain previously characterized taxa, this can be used to automatically compare new genomes and MAGs with hundreds or thousands of phylogenetically related genome sequences (**Fig. 3**).

5. When it comes to comparing phylogenetic trees, the authors should refer to standardized metrics to assess the dissimilarity of phylogenetic trees, e.g. the quartet distances, rather than making statements such as ‘PhyloPhlAn2’s tree was very consistent relative to this previous manually curated phylogeny (...)’.

We assessed differences in tree topology using the tqDist (Sand et al. 2014) function available in the R package “quartet” and found that there was a 90.5% consistency between the *Staphylococcus aureus* phylogenies. We have added this information to the text and updated the **Supplementary Fig. 2C**. We have also rephrased the statement and added the corresponding text to the methods section:

When comparing PhyloPhlAn2’s tree with the previous manually curated phylogeny, we found an overall correlation of 0.992 (Pearson’s correlation) between normalized pairwise branch length distances from the two phylogenies (**Fig. 2B**) and 90.5% consistency between quartet distances (**Supplementary Fig. 2**).

Staphylococcus aureus and Escherichia coli MLST and phylogroup analyses

We used PhyloPhlAn2 to generate the phylogenies of 1000 *S. aureus* reference genomes and 135 *S. aureus* isolates as discussed in the results. To evaluate the phylogeny generated by PhyloPhlAn2 we used the tqDist (Sand et al. 2014) function available in the R *quartet* package to compare quartet distances between the PhyloPhlAn2 and the manually curated reference phylogeny (Manara et al., n.d.).

Supplementary Figure S2. Side-by-side comparison of *Staphylococcus aureus* phylogenies. (A) The original phylogeny manually curated (Manara et al. 2018). (B) The phylogeny automatically reconstructed by PhyloPhlAn2 and visualized with GraPhlAn using the same annotations of (A). (C) Ternary plot of quartet distances showing the consistency between *Staphylococcus aureus* phylogenies.

6. I was a bit confused whether 2,127 core genes (figure caption) or 1,658 core genes, which the main text appears to indicate, were used to reconstruct the phylogeny shown in Figure 2A. This should be clarified.

We thank the reviewer for pointing out this, which has now been clarified in the text. Briefly, the 2,127 are the total number of UniRef90 core proteins present in the database PhyloPhlAn2 retrieved for *S. aureus*, while the 1,658 are the actual number of UniRef90 used in the phylogenetic analysis. This is due to the fact that PhyloPhlAn can compute a set of core genes/proteins based on the current inputs provided and according to the value specified for the "--min_num_entries" parameter, which will retain for the phylogeny reconstruction only those genes/proteins with at least that number of input genomes/proteomes, and discard all genes/proteins not meeting that value.

7. In the caption of figure 4, the authors refer to a sequence alignment comprising 4,522 amino acid loci. The use of 'loci' here should be avoided.

We thank the reviewer for pointing out this, the text has been corrected:

Figure 4. PhyloPhlAn2 efficiently scales to provide a microbial tree of life including 17,672 species-representative genomes spanning 51 known phyla and 84 additional candidate phyla. With 17,672 species-dereplicated isolate genomes and MAGs as input (see **Methods**), PhyloPhlAn2 used 400 optimized universal marker sequences to produce a pan-microbial phylogeny in approximately 10 days (~24,000 CPU-hours on 100 parallel cores). The underlying multiple sequence alignment comprised 4,522 amino acid positions from among 1,872,710 in the untrimmed concatenated marker alignments.

8. I do not really understand the distinction between MAGs and genome sequences obtained from bacterial isolates which is made repeatedly in the text. To my understanding, both represent genome sequences representing individual taxa. Why the would one want to make this difference?

We agree that both MAGs and genomes from bacterial isolates represent the same biological entity. However, MAGs have a set of potential associated problems ranging from limited completeness to contaminant contigs/SNPs, and many researchers in the field argued that this distinction should always be made clear whenever possible.

9. In the example analysis, the authors reduce the number of taxa for the phylogenetic tree reconstruction by selecting for each species one representative genome. They then test whether the quality of the selected genomes is sufficient to have them included in the tree reconstruction. This results in the exclusion of about 2,000 taxa. I am wondering whether for these excluded 2,000 taxa there is no other representative available that would meet the quality criteria. In essence, why was the quality screen performed after the 'dereplication' and not beforehand?

We thank the reviewer for this comment. The quality control described in the text was showing the fact that PhyloPhlAn2 internally can screen the provided input genomes, to select those that from a phylogenetic point of view map a minimum number of (in this case) universal markers. Because of this, we did not know prior running PhyloPhlAn2 that the ~2,000 genomes did not map a sufficient number of universal markers (threshold set to 100 markers out of the 400 in the PhyloPhlAn database (25% of the total)), as they were already selected as representative, such as those maximizing completeness and minimizing contamination as described in “Methos details / Grouping of metagenomic assemblies into species-level genome bins” in Pasolli et al., Cell, 2019.

We amend the text as follows to clarify this:

These were dereplicated to one representative per species by hierarchical clustering on genomic distances as estimated by Mash (Ondov et al. 2016) with cluster cutoff at 5% intra-cluster nucleotide identity (see **Methods**), resulting in 19,607 clusters. Additional automatic quality control available in PhyloPhlAn2, removed genomes containing less than 100 of PhyloPhlAn2’s 400 optimized deep-branching marker genes (**Methods**), resulting in 17,672 representative genomes in the final tree

10. In the methods, the authors refer to high-scoring alignment positions and consider them as ‘phylogenetically relevant’. What does this term mean exactly? And how confident are the authors that the scores indeed are linked to something that I would like to call phylogenetic informativeness of an alignment column? For example, I am unaware that Muscle has implemented anything in this direction.

We thank the reviewer for pointing out this. MUSCLE indeed has an option “-scorefile” which outputs an extra output file that provides a score for each column in the MSA. This scoring simply computes for all columns the sum of all pairwise scores based on the VTML200 substitution matrix. The scoring functions implemented in PhyloPhlAn2 are based on the work of (Valdar 2002) were, on a smaller scale, several scoring functions have been tested. Also, in the back-to-back work from (Zhu et al. 2019), we validated and shown in both **Supplementary Fig. 6** (this work) and **Fig. 3, Supplementary Fig. 11, 12, 17, Supplementary Note 2 and 5** (Zhu et al. 2019) the effect of different sampling methods implemented in PhyloPhlAn2.

This has been now added explicitly in the text:

The concatenated MSA contained 4,522 amino acids out of 1.87M of total length of the untrimmed concatenated marker sequence alignments. The selection of these most phylogenetically informative positions in the MSA is performed by PhyloPhlAn2 in this aggressive setting for scalability purposes and was validated as we reported elsewhere (Zhu et al. 2019) using the “trident” scoring function (Valdar 2002).

11. In the manuscript, and here in particular with respect to Figs. 2 and 3, it should be clearly specified which analyses have been performed with PhyloPhlAn2, and which have been made separately. If the results shown in figures 2 and 3 have not been generated directly with PhyloPhlAn2, then this should be explicitly stated. Then the authors should explain in the supplementary material in detail how the input data that is required for these analyses can be extracted from PhyloPhlAn2. If they have all been created with this package, then it would be great to see an example workflow in the supplement.

We have edited figure captions for Figures 2 and 3 to clearly specify that these analyses were performed using PhyloPhlAn2 outputs. For each example analysis shown in the manuscript using PhyloPhlAn2 there is a corresponding wiki page with the workflow used to produce most of the analysis shown in figures 2-4. These can be accessed using the following links:

1. Phylogenetically characterized isolate genomes of a given species (*S. aureus*)
2. Tree of life
3. Metagenomics
4. High-resolution phylogeny of known genomes reconstructed from metagenomes of a given species (*E. coli*)
5. Phylogenetically characterization of unknown SGB from Proteobacteria phylum

Reviewer #3 (Remarks to the Author):

The paper by Asnicar et al describes the PhyloPhlan2 software. While we believe the tool is good quality and does important things, we find that the paper (and software) needs some substantial work.

Major and minor points in the paper:

* Nothing serves to unite the family-focused view and the strain-focused view.

PhyloPhlan2 currently offers two levels of phylogenetic marker resolution; a set of 400 universal markers and species-specific markers that are downloaded and created based on pangenomes of UniRef90 protein clusters. Computing phylogenetic markers of different taxonomic levels and resolutions fall outside the scope of the framework, as these will likely change on a case-by-case basis. The framework, however, allows the use of a custom set of phylogenetic markers that can better serve a user's desired phylogenetic resolution.

We have made this information clearer in the methods section of text:

PhyloPhlan2 can also consider any set of markers computed by the user with different strategies and provided as a fasta sequence file for either amino acids or nucleotides. These markers can be at a higher or lower resolution as those currently provided by the framework and can be integrated using the database setup script (`phylophlan2_setup_database.py`).

* The software has only been used on human microbiome samples so far, we believe? How well does it extend to non-human data?

We thank the reviewer for this question and we noticed that this point was not clear from the original writing of the manuscript. Indeed the microbial tree of life proposed in **Fig. 4** contains also non-human-related genomes as we included 11,402 kSGBs (**Fig. 1B** from (Pasolli et al. 2019)) for which we never reconstructed a MAG from a human microbiome sample (Pasolli et al. 2019) and also genomes reconstructed from the work of (Parks et al. 2017), which include not only genomes reconstructed from human gut metagenomes, but also from rumen, guinea pigs, and baboon faeces. To better highlight this we are providing a new supplementary figure that highlights only the non-human MAGs in the tree of life (**Supplementary Fig. 5**).

Supplementary Figure 5. PhyloPhlAn2 tree-of-life highlighting non-human SGBs. The phylogeny is the same as shown in Fig. 4, but highlights such SGBs in which there is no present a MAG reconstructed from a human sample.

We also add a comment about this in the text as reported below:

While Proteobacteria are prevalently found in non-human samples, Actinobacteria are instead mainly associated with human samples. Firmicutes and Bacteroidetes are more equally derived from both human and non-human samples, with some preferences in specific sub-trees of the two phyla (**Supplementary Fig. 5**).

* The software seems to package many tools that do somewhat overlapping things, which confuses the message in the paper. It would be good to be really clear and explicit about what software is being used for which functionality.

We thank the reviewer for their comment and would like to point out that we improved the **Supplementary Table 1** that better shows all differences between the first version of PhyloPhlAn and PhyloPhlAn2. The table also highlights the different functionalities and the software that is used in each step. For most of the steps, the choice of the tool to use can be defined by the user, such as for the multiple sequence alignment, MSA position scoring, and

phylogeny inference. For tools that have overlapping functions and cannot be altered by the user, we have included additional information in the table by expliciting what each software is used for.

We also modified the text that now reads:

Most of these features are unique to PhyloPhlAn2 and were not available in the first version of the framework as detailed in the comparison table **Supplementary Table 1**.

* What is the point of and value of taxonomy here? The paper is largely about assigning taxonomy to MAGs, and it would be good to have a sentence or two about why this is important.

We thank the reviewer for pointing out this and we agree that this can be made more explicit in the text, in addition to the following sentence that was already present in the Introduction:

Phylogenetic and corresponding taxonomic characterization is crucial in microbial genomics, for contextualizing new genomes without prior phenotypic information, and for determining their genetic novelty and genotype-phenotype relationships.

We edited the text in the “**Robust taxonomy assignment for metagenome-assembled genomes using species-level genome bins**” that now reads as follows:

If needed, this procedure is repeated for higher taxonomic clades with genus-level genome bins (GGBs, up to 15% genomic distance) and family-level genome bins (FGBs, up to 30% genomic distance), ultimately providing a more comprehensive taxonomic context for the set of input genomes that is crucial to guide downstream analyses and complement their phylogenetic placement.

* MAGs are not reference genomes; the two are often conflated in the paper.

We thank the reviewer for pointing out this inconsistency, we revised and updated the text to make explicit whether we are referring to reference genomes and/or MAGs.

* It is unclear why the Ethiopian data was included as a test data set. Is there any motivation beyond wanting to provide an example with unpublished data?

We apologize if this was unclear, but we chose to include the Ethiopian MAGs as test data because they came from non-westernized individuals and present a high probability of possessing unseen microbial diversity, which is particularly relevant for phylogenetic analysis in general and to demonstrate how PhyloPhlAn2 can be used for both taxonomic

classification using Mash distances and the SGB database and for further phylogenetic placement and characterization of these newly recovered genomes. We could have used, in principle, other public metagenomes but because our reference set of MAGs and reference genomes from (Pasolli et al. 2019) includes already a large fraction of human-associated metagenomic data, we preferred to use a novel set of samples that was not already included in the database.

We have modified the text in order to be clearer why the Ethiopian data was used as an example analysis:

We used PhyloPhlAn2 to taxonomically place a set of MAGs retrieved from a cohort of 50 rural Ethiopian individuals (see **Data availability**) only used so far to characterize *Prevotella copri* strains (Tett et al. 2019), as these samples had not been used in the generation of SGBs and are likely to contain substantial unseen phylogenetic diversity.

Software questions:

There is little to no detail provided on the software requirements -- much or all of the below should be addressed somewhere.

* what are the basic memory and compute requirements?

We thank the reviewer for pointing out the lack of details in the original manuscript about the computational requirements of the tools. We did not perform a thorough benchmark analysis about memory requirements, as this would mean testing all software available that can perform the internal steps of PhyloPhlAn2. The memory usage is heavily dependent on both the external tools used and the size of the markers database, so this will make impractical to have a general estimation. The minimal compute requirements is one core. Partially, we answered this in Question 9 from Reviewer #2, for which we report below the edited text:

PhyloPhlAn2 modularity allows to parallelize internally to the framework the steps that are independent and can execute in parallel. Otherwise, PhyloPhlAn2 pass on the available number of cores specified by the user to the single program that can exploit the multi-processing computation.

* how do you modify parameters, if at all?

We thank the reviewer for pointing out the lack of details in the original manuscript about the details on personalization of the pipeline and computational requirements of the tools. We answered this in the Questions 1 and 2 from Reviewer #2.

Amended text now reads:

Each tool can be separately and sequentially applied providing full step-by-step control on the whole phylogenetic analysis, but doing so requires substantial expertise not only in identifying the right targets, parameters, and steps for computational phylogenetics, but also in understanding how such tools should be interfaced one with the other.

Comparison of automatically obtained phylogenies with respect to manually curated and evaluated phylogenetic trees showed that PhyloPhlAn2 is highly accurate at different resolutions, ranging from species-level clades to the whole tree-of-life. While for several tasks the fully automatic pipeline should already provide the answer for the problem at hand, our pipeline permits extensive customization of each step for more *in-depth* and personalized analyses. Therefore, we anticipate that PhyloPhlAn2 will serve as a useful instrument to understand present and future microbial diversity in a wide range of microbiological and ecological settings.

Integration of new tools not available in the different steps of the framework can be achieved by manually editing the configuration files and inserting the desired tools/parameters, as long as input and output files are in the same format of currently implemented tools. This procedure is described with a dedicated section (“Integrating new tools in the framework”) in the documentation available in the PhyloPhlAn2 code repository.

* how is traceability and provenance implemented?

We thank the reviewer for pointing out the lack of details about this aspect. The PhyloPhlAn2 way of dealing with traceability and provenance is through both the configuration file and the output log provided during the analysis. The config file contains all the details about the external tools used for performing the phylogenetic analysis, while the output log contains detailed information about the execution of the several different steps performed. Taken together with the output log, can be used to make the obtained results reproducible.

We added this information in the Methods:

The PhyloPhlAn2 pipeline relies on both the configuration file and the output log generated during the analysis to track which external tools have been used with their specific set of parameters and the details of the execution, to make the obtained results reproducible.

* is the software open source? under what license? this should be mentioned in the paper.

PhyloPhlAn2 is released and distributed open-source under the MIT license. We reported these information in the **Data availability** section in the **Methods** and in the “license.txt” file present in the repository. The **Data availability** section is reported below for clarity:

Data availability

PhyloPhlAn2 is released open-source and available in Bitbucket at <https://bitbucket.org/nsegata/phylophlan> and the current version used in this work is version 2.0. Manuals and online tutorials describing the PhyloPhlAn2 framework are available at <https://bitbucket.org/nsegata/phylophlan/wiki/phylophlan2>. User support is provided both through the issues tracking system in the Bitbucket repository (<https://bitbucket.org/nsegata/phylophlan/issues>) and the Support Group phylophlan-users@googlegroups.com. Raw metagenomes for the Ethiopian cohort are available under NCBI-SRA BioProject id PRJNA504891 and the 369 MAGs can be downloaded from the software page at <http://segatalab.cibio.unitn.it/tools/phylophlan2>.

* how is the software packaged?

The PhyloPhlAn2 software is now available through the Bitbucket repository with a conda environment that takes care of the required python packages. We plan to release the major updates of PhyloPhlAn2 also as conda package in Bioconda.

* what is the approach, if any, to automated software testing?

We do not have an automatic testing software, we use a set of examples to evaluate whether the pipeline after having done the needed changes still produces results comparable to the previous version, or more accurate, depending on the type of edit.

* what is the approach, if any, to versioning?

We do not have an automatic way to tag specific versions of the pipeline. Major versions of the software will be tagged manually on the Bitbucket repository and will be packaged and made available in the Bioconda system.

* what are the scalability aspects of the software? does it parallelize nicely?

We thank the reviewer for pointing out the lack of details about this aspect in the original manuscript. The same question was raised in Question 9 by Reviewer #2, and we provided an answer above.

The edited text in the manuscript now reads:

PhyloPhlAn2 modularity allows to parallelize internally to the framework the steps that are independent and can be executed in parallel. Otherwise, PhyloPhlAn2 pass on the available number of cores specified by the user to the single program that can then internally exploit the multi-processing computation.

* How big is the database on disk, how do you update the database, etc?

We thank the reviewer for pointing out this aspect. A similar question was raised in the Question 5 by Reviewer #2, and we provided an answer above.

The added section to the Methods is reported below:

PhyloPhlAn2 databases management

Several convenience scripts are available in PhyloPhlAn2 to handle the databases at different scales and for different analyses. In particular, the scripts `phylophlan2_get_reference.py`, `phylophlan2_setup_database.py`, and `phylophlan2_metagenomic.py` have been developed to handle different database files that are (i) automatically retrieved when needed and only if not present locally, (ii) stored locally after the download, and (iii) updated when the users specify the “`--database_update`” parameter. Database files comprise the set of pre-computed species-specific set of UniRef90 proteins, the set of available genomes from GenBank, and the SGB release.

The tutorials on the wiki don't seem to work for us.

We apologize if these tutorials were not working before and thank the reviewer for pointing this out. We have now re-checked them and were able to reproduce the analysis. These can be accessed by the main PhyloPhlAn2 tutorial (<https://bitbucket.org/nsegata/phylophlan/wiki/phylophlan2>) and can be directly accessed using the following links:

1. Phylogenetically characterized isolate genomes of a given species (*S. aureus*)
2. Tree of life
3. Metagenomics
4. High-resolution phylogeny of known genomes reconstructed from metagenomes of a given species (*E. coli*)
5. Phylogenetically characterization of unknown SGB from the Proteobacteria phylum

The files and pipelines mentioned in the paper do not seem to be available.

We again apologize and thank the reviewer for pointing this out. We have now re-checked and re-tested them and we were able to reproduce the analyses from scratch.

References for the response letter

- Asnicar, Francesco, George Weingart, Timothy L. Tickle, Curtis Huttenhower, and Nicola Segata. 2015. "Compact Graphical Representation of Phylogenetic Data and Metadata with GraPhlAn." Edited by Jaume Bacardit. *PeerJ* 3 (June): e1029.
- Bursteinas, Borissas, Ramona Britto, Benoit Bely, Andrea Auchincloss, Catherine Rivoire, Nicole Redaschi, Claire O'Donovan, and Maria Jesus Martin. 2016. "Minimizing Proteome Redundancy in the UniProt Knowledgebase." *Database: The Journal of Biological Databases and Curation* 2016 (December).
<https://doi.org/10.1093/database/baw139>.
- David, Lawrence A., Ana Weil, Edward T. Ryan, Stephen B. Calderwood, Jason B. Harris, Fahima Chowdhury, Yasmin Begum, Firdausi Qadri, Regina C. LaRocque, and Peter J. Turnbaugh. 2015. "Gut Microbial Succession Follows Acute Secretory Diarrhea in Humans." *mBio* 6 (3): e00381–15.
- Maiden, M. C., J. A. Bygraves, E. Feil, G. Morelli, J. E. Russell, R. Urwin, Q. Zhang, et al. 1998. "Multilocus Sequence Typing: A Portable Approach to the Identification of Clones within Populations of Pathogenic Microorganisms." *Proceedings of the National Academy of Sciences of the United States of America* 95 (6): 3140–45.
- Manara, Serena, Francesco Asnicar, Francesco Beghini, Davide Bazzani, Fabio Cumbo, Moreno Zolfo, Eleonora Nigro, et al. n.d. "Microbial Genomes from Gut Metagenomes of Non-Human Primates Expand the Primate-Associated Bacterial Tree-of-Life with over 1,000 Novel Species." *Genome Biology*.
- Manara, Serena, Edoardo Pasolli, Daniela Dolce, Novella Ravenni, Silvia Campana, Federica Armanini, Francesco Asnicar, et al. 2018. "Whole-Genome Epidemiology, Characterisation, and Phylogenetic Reconstruction of Staphylococcus Aureus Strains in a Paediatric Hospital." *Genome Medicine* 10 (1): 82.
- Nguyen, Lam-Tung, Heiko A. Schmidt, Arndt von Haeseler, and Bui Quang Minh. 2015. "IQ-TREE: A Fast and Effective Stochastic Algorithm for Estimating Maximum-Likelihood Phylogenies." *Molecular Biology and Evolution* 32 (1): 268–74.
- Obregon-Tito, Alexandra J., Raul Y. Tito, Jessica Metcalf, Krithivasan Sankaranarayanan, Jose C. Clemente, Luke K. Ursell, Zhenjiang Zech Xu, et al. 2015. "Subsistence Strategies in Traditional Societies Distinguish Gut Microbiomes." *Nature Communications* 6 (March): 6505.
- Ondov, Brian D., Todd J. Treangen, Páll Melsted, Adam B. Mallonee, Nicholas H. Bergman, Sergey Koren, and Adam M. Phillippy. 2016. "Mash: Fast Genome and Metagenome Distance Estimation Using MinHash." *Genome Biology* 17 (1): 132.
- Parks, Donovan H., Maria Chuvochina, David W. Waite, Christian Rinke, Adam Skarszewski, Pierre-Alain Chaumeil, and Philip Hugenholtz. 2018. "A Standardized Bacterial Taxonomy Based on Genome Phylogeny Substantially Revises the Tree of Life." *Nature Biotechnology* 36 (10): 996–1004.
- Parks, Donovan H., Christian Rinke, Maria Chuvochina, Pierre-Alain Chaumeil, Ben J. Woodcroft, Paul N. Evans, Philip Hugenholtz, and Gene W. Tyson. 2017. "Recovery of Nearly 8,000 Metagenome-Assembled Genomes Substantially Expands the Tree of Life." *Nature Microbiology* 2 (11): 1533–42.
- Pasolli, Edoardo, Francesco Asnicar, Serena Manara, Moreno Zolfo, Nicolai Karcher, Federica Armanini, Francesco Beghini, et al. 2019. "Extensive Unexplored Human Microbiome Diversity Revealed by Over 150,000 Genomes from Metagenomes Spanning Age, Geography, and Lifestyle." *Cell* 176 (3): 649–62.e20.
- Rampelli, Simone, Stephanie L. Schnorr, Clarissa Consolandi, Silvia Turrone, Marco Severgnini, Clelia Peano, Patrizia Brigidi, Alyssa N. Crittenden, Amanda G. Henry, and Marco Candela. 2015. "Metagenome Sequencing of the Hadza Hunter-Gatherer Gut

- Microbiota." *Current Biology: CB* 25 (13): 1682–93.
- Sand, Andreas, Morten K. Holt, Jens Johansen, Gerth Stølting Brodal, Thomas Mailund, and Christian N. S. Pedersen. 2014. "tqDist: A Library for Computing the Quartet and Triplet Distances between Binary or General Trees." *Bioinformatics* 30 (14): 2079–80.
- Segata, Nicola, Daniela Börnigen, Xochitl C. Morgan, and Curtis Huttenhower. 2013. "PhyloPhlAn Is a New Method for Improved Phylogenetic and Taxonomic Placement of Microbes." *Nature Communications* 4: 2304.
- Suzek, Baris E., Hongzhan Huang, Peter McGarvey, Raja Mazumder, and Cathy H. Wu. 2007. "UniRef: Comprehensive and Non-Redundant UniProt Reference Clusters." *Bioinformatics* 23 (10): 1282–88.
- Tett, Adrian, Kun D. Huang, Francesco Asnicar, Hannah Fehlner-Peach, Edoardo Pasolli, Nicolai Karcher, Federica Armanini, et al. 2019. "The Prevotella Copri Complex Comprises Four Distinct Clades Underrepresented in Westernized Populations." *Cell Host & Microbe* 26 (5): 666–79.e7.
- UniProt Consortium. 2014. "Activities at the Universal Protein Resource (UniProt)." *Nucleic Acids Research* 42 (Database issue): D191–98.
- Valdar, William S. J. 2002. "Scoring Residue Conservation." *Proteins* 48 (2): 227–41.
- Zhu, Qiyun, Uyen Mai, Wayne Pfeiffer, Stefan Janssen, Francesco Asnicar, Jon G. Sanders, Pedro Belda-Ferre, et al. 2019. "Phylogenomics of 10,575 Genomes Reveals Evolutionary Proximity between Domains Bacteria and Archaea." *Nature Communications* 10 (1): 5477.
- Zolfo, Moreno, Adrian Tett, Olivier Jousson, Claudio Donati, and Nicola Segata. 2017. "MetaMLST: Multi-Locus Strain-Level Bacterial Typing from Metagenomic Samples." *Nucleic Acids Research* 45 (2): e7.

Reviewers' comments:

Reviewer #1 (Remarks to the Author):

I thank the authors for their responses to my previous concerns. However, there are still aspects of the methodology used by PhyloPhlAn 2 for taxonomic assignments which I do not find to be clearly articulated in the manuscript.

1. I still find the section related to the assignment of query genomes to SGBs, GGBs and FGBs unclear. Perhaps this is primarily a matter of terminology. What is a GGB? This is not defined in the manuscript beyond "genus-level genome bins". Does this mean all sequences within a genus were combined into a single FASTA file? I presume not. My guess is that a query genome is only ever compared to PhyloPhlAn reference genomes and that a query genome might be assigned to the genus or family of the closest identified SGB (or set of closest SGBs). Is this correct? Notably, the concept of GGBs and FGBs is not detailed in the "Pipeline for taxonomic assignment of genomes and MAGs" method section which I believe should be expanded to make it clear how taxonomic labels above the rank of species are assigned.

2. It was indicated that the 15% ANI threshold for GGBs was justified in Pasolli et al., 2019. This citation should be added to the sentence indicating 15% was used for GGBs. Does Pasolli et al., 2019 also justify the use of 30% genomic distance for FGBs? What happens above the rank of family?

3. In your response to question 3 in my last review, you indicated that the MAGs were assigned to the Chlamydiae phylum based on their average Mash distance to the SGB resource. Related to point 1, it remains unclear to me what criteria is used to assert a taxonomic assignment at the phylum level as opposed to a more specific rank (e.g., class, order, ...). Is there an ANI cutoff used to assert that a classification can only be resolved to the phylum level or is an LCA approach across a set of closest SGBs used?

Reviewer #2 (Remarks to the Author):

I am happy to say that the authors did a nice job in addressing the reviewers' comments. Only few and considerably minor points remain.

1. By stating that the data confirms that the ethopian uSGB (ID 19436) represents a new species and genus, the authors either imply that the genus *Campylobacter* is paraphyletic, or the three shown in figure 3C is wrong. Either is possible, and I think this should be briefly discussed. Note in this context that a bootstrap support of 80 is not sufficient evidence that the tree is true. There is a long list of literature discussing how modelling artefacts in the course of phylogeny reconstruction can come to highly supported but wrong trees.

2. I am missing a scale bar in Supplementary Figure 4

3. I appreciate that the authors added the quartet distance as a measure to compare the trees shown in Suppler Fig 2A and B. I am just wondering to what extent the result of 90.5% consistence between quartet distances the authors report is driven by the many quartets that connect the very closely related strains. If I look at the two phylogenies, I see quite some differences in the topology within the clade making up the right part of the tree. I use the numbers next to the individual sub-clades as labels to clarify my point: Clade 97 is placed as sister to (772,(5,228)) in the curated phylogeny, while it is placed differently in the PhyloPhlAn2 tree. Likewise, the clade 88 is placed differently, as are (25,96). In essence, on a closer look it appears that there quite some differences in the tree topology, and I have the feeling that they are somehow watered down. I suggest to mention these differences and to then explain - if this is true - that the differences are caused by rather poorly supported splits in the trees.

4. I am somehow not really happy with the recurrent use of tree-of-life. For example, what is a 'tree-of-life'-size phylogeny? Moreover, throughout your analysis, you are showing only trees without eukaryotes. Thus, it is only the bacterial and archaeal section of the TOL. I leave it to the discretion of the authors to solve this issue.

Ingo Ebersberger

Reviewer #3 (Remarks to the Author):

The paper "Precise phylogenetic analysis of microbial isolates and genomes from metagenomes using PhyloPhlAn2" describes a significant update to the PhyloPhlAn software. This is a review of an initial revision to the paper.

I appreciate the authors' responses to the reviewers, and the revised paper reads quite well. With respect to my previous review, I could not make time to attempt to rerun the software for this revision, my apologies.

I have a few relatively minor suggestions for revision, and one major concern.

My major concern stems from the fact that the NCBI taxonomy is increasingly problematic for classifying genomic content in at least some areas of the taxonomic tree; this is what (I believe) spurred the development of the GTDB taxonomy, which seeks to eliminate these inconsistencies. Given confusion around taxonomic assignments as well as a low but steady rate of contamination in newly incorporated MAGs, it is likely that there will be challenges ahead for any taxonomic software in classification if they use the NCBI taxonomy. So my major concern is this: how does PhyloPhlAn2 deal with confused NCBI taxonomy?

For example, we have found that phyla Actinobacteria and Acidobacteria contain species-level genomes that are nearly identical between the two phyla. The same is true of Actinobacteria and Chloroflexi, and Chlamydiae and Firmicutes, among others.

We also find that GTDB-Tk (which is marker based) and ANI-based approaches yield quite different answers for many MAGs, which highlights the challenges facing the field. (We do not yet understand why; this is ongoing work.)

I note that in the authors' response to Reviewer #1, they highlight a potential taxonomic assignment inconsistency. It might be worth making a point about this in the paper, and or providing a flag for this in the output of PhyloPhlAn2.

It is clearly not reasonable to require PhyloPhlAn2 to fix NCBI's classification, but I do think it is important for PhyloPhlAn2 to alert the user in situations where the NCBI taxonomy may be problematic and inconsistent for the results reported by PhyloPhlAn2.

Actually, reading through the new text "PhyloPhlan2 can scale to microbial tree of life reconstructions..." it seems like the database preparation step may simply make arbitrary choices here (based around dereplicating genomes on 95% ANI). If that's the case, then the software may not be able to flag confusing situations, and that should be noted explicitly.

Minor suggestions --

Line 223, the Zou paper used EzBioCloud to assign taxonomic identity, it seems. I'm not sure about the use of the word "accuracy" in that sentence; perhaps consistency would be a better word? (This is a soft suggestion, I leave it up to the authors.)

Please provide a version-specific DOI for the software used in this publication.

We thank all the reviewers for their comments as they helped improve the clarity of our manuscript. We report below the point-by-point answer to all comments. The reviewers will notice that we changed the versioning system for the software which is now referred to as “PhyloPhlAn 3.0” (instead of “PhyloPhlAn2”). This was necessary because this new version of PhyloPhlAn will be part of the bioBakery 3.0 software suite and so the name of the tools have been updated to be in sync with each other. This version of PhyloPhlAn was not publicly advertised yet, and so changing the version number at this point is safe for the users and will guarantee better compatibility with related software.

Reviewer #1 (Remarks to the Author)

I thank the authors for their responses to my previous concerns. However, there are still aspects of the methodology used by PhyloPhlAn 2 for taxonomic assignments which I do not find to be clearly articulated in the manuscript.

1. I still find the section related to the assignment of query genomes to SGBs, GGBs and FGBs unclear. Perhaps this is primarily a matter of terminology. What is a GGB? This is not defined in the manuscript beyond “genus-level genome bins”. Does this mean all sequences within a genus were combined into a single FASTA file? I presume not. My guess is that a query genome is only ever compared to PhyloPhlAn reference genomes and that a query genome might be assigned to the genus or family of the closest identified SGB (or set of closest SGBs). Is this correct? Notably, the concept of GGBs and FGBs is not detailed in the “Pipeline for taxonomic assignment of genomes and MAGs” method section which I believe should be expanded to make it clear how taxonomic labels above the rank of species are assigned.

We agree with the Reviewer that a more extensive definition of GGBs (and FGBs) is needed. According to Pasolli et al. 2019, GGBs are defined as bins of genomes with an average all-vs-all Mash genetic distance <15%. Briefly, following the same approach used to define SGBs at <5% Mash genetic distance, the 15% threshold for GGBs was chosen as a result of the minimization of genus-level under-clustering (i.e. genomes belonging to different genera falling into the same GGB) and genus-level over-clustering (i.e. genomes belonging to the same genus falling in two different GGBs). This analysis is reported in Figure S2 of Pasolli et al., Cell, 2019 and discussed in the methods and the results. Overall, this means that a GGB is a proxy for a bacterial genus. The same approach was used for defining FGBs, using a genetic distance threshold of 30% that minimizes the under- and over-clustering errors at the family taxonomic level.

We moreover thank the Reviewer for pointing out that it was not clear how taxonomic labels are assigned to GGBs and FGBs. Applying the same approach we use to assign a taxonomic label to each SGB, to each GGB we assign the genus label of the closest reference genomes that have a genetic distance <15%, and to each FGB the family label of the closest reference genomes with a genetic distance <30%. In the case more than one reference genomes are present in the GGB (and FGB) and they have different taxonomic labels at the genus (or family) level, a majority voting rule is used to assign the genus (or family) taxonomic label.

We modified the text in the “Pipeline for taxonomic assignment of genomes and MAGs” section in the Methods to reflect these clarifications (and we better refer to this section from the main text when we mention the GGBs):

For the cases in which PhyloPhlAn 3.0 cannot assign an SGB to an input genome, the assignment procedure is repeated at the level of genus-level genome bins (GGBs) and family-level genome bins (FGBs). Similarly to SGBs, GGBs and FGBs were defined elsewhere ⁴¹ via hierarchical average linkage clustering at 15% and 30% genetic distance, respectively. These thresholds were empirically estimated in the same work as those more closely reflecting the genetic span of the known taxonomically defined genera and families. GGBs and FGBs are also taxonomically assigned to known genus and family labels if the clusters comprise one or more reference genomes within the corresponding average genetic distance (15% for GGBs, 30% for FGBs, in the case of taxonomic inconsistencies in reference genomes falling inside the same SGB/GGB/FGB, a majority voting approach is applied to assign the most represented taxonomic label). Using this definition of GGBs and FGBs, PhyloPhlAn 3.0 assigns input genomes missing SGB assignment (i.e. the input genome is at >5% average genetic distance with respect to all SGBs) to the closest GGB and/or FGB that are at an average genetic distance <15% and <30%, respectively. If the average genetic distance of the input genome is >30% to any FGBs, limitations in nucleotide similarity quantification methods would not allow reliable higher-level taxonomic assignment ⁴¹. In these cases, PhyloPhlAn 3.0 reports the phylum label of the set of closest reference genomes (i.e. the set of genomes within 5% genetic distance from the closest) decided via majority voting.

2. It was indicated that the 15% ANI threshold for GGBs was justified in Pasolli et al., 2019. This citation should be added to the sentence indicating 15% was used for GGBs. Does Pasolli et al., 2019 also justify the use of 30% genomic distance for FGBs? What happens above the rank of family?

Similarly to what is reported in the answer to Question 1, in Pasolli et al. 2019 the 30% genomic distance was identified as the threshold minimizing the under- (i.e. genomes belonging to different families being assigned to the same FGB) and over-clustering errors (i.e. genomes belonging to the same family being assigned to different FGBs), meaning that an FGB is a proxy for a bacterial family, with genomes belonging to a specific FGB showing an all-vs-all average Mash genetic distance <30%.

At taxonomic levels above the family rank, there is no reliable method that can accurately estimate the genetic divergence, therefore no threshold can be set for the definition of higher taxonomic levels, such as classes, orders, or phyla. Because of this, for those SGBs assigned to a GGB and FGB lacking a taxonomic label, we first identify their closest reference genome and those reference genomes that are within 5% genomic distance from the closest. We then report the phylum taxonomic label associated with them. In the case of

non-consistent taxonomies among these reference genomes, a majority voting rule is applied to select the phylum taxonomic label.

To include this information, we modified the text as follows in the “Phylogenetic context for taxonomically assigned and unassigned input genomes” section:

In the Ethiopian study, we focused on the prevalent human gut colonizer *Escherichia coli* (kSGB ID 10068), and on the most prevalent uSGB (ID 19436, 13 MAGs in total) for which the closest reference genomes belonged to the Proteobacteria phylum.

We moreover added the citation as suggested by the Reviewer:

If needed, this procedure is repeated for higher taxonomic clades with genus-level genome bins (GGBs, up to 15% genomic distance) and family-level genome bins (FGBs, up to 30% genomic distance, see **Methods**) (Pasolli et al. 2019), ultimately providing a more comprehensive taxonomic context for the set of input genomes to guide downstream analyses and complement their phylogenetic placement.

Additionally, we updated the SGB, GGB, and FGB assignment procedures in the Methods (sub-section “Pipeline for taxonomic assignment of genomes and MAGs”) as already reported above:

For the cases in which PhyloPhlAn 3.0 cannot assign an SGB to an input genome, the assignment procedure is repeated at the level of genus-level genome bins (GGBs) and family-level genome bins (FGBs). Similarly to SGBs, GGBs and FGBs were defined elsewhere⁴¹ via hierarchical average linkage clustering at 15% and 30% genetic distance, respectively. These thresholds were empirically estimated in the same work as those more closely reflecting the genetic span of the known taxonomically defined genera and families. GGBs and FGBs are also taxonomically assigned to known genus and family labels if the clusters comprise one or more reference genomes within the corresponding average genetic distance (15% for GGBs, 30% for FGBs, in the case of taxonomic inconsistencies in reference genomes falling inside the same SGB/GGB/FGB, a majority voting approach is applied to assign the most represented taxonomic label). Using this definition of GGBs and FGBs, PhyloPhlAn 3.0 assigns input genomes missing SGB assignment (i.e. the input genome is at >5% average genetic distance with respect to all SGBs) to the closest GGB and/or FGB that are at an average genetic distance <15% and <30%, respectively. If the average genetic distance of the input genome is >30% to any FGBs, limitations in nucleotide similarity quantification methods would not allow reliable higher-level taxonomic assignment⁴¹. In these cases, PhyloPhlAn 3.0 reports the phylum label of the set of closest reference genomes (i.e. the set of genomes within 5% genetic distance from the closest) decided via majority voting.

3. In your response to question 3 in my last review, you indicated that the MAGs were assigned to the Chlamydiae phylum based on their average Mash distance to the SGB resource. Related to point 1, it remains unclear to me what criteria is used to assert a

taxonomic assignment at the phylum level as opposed to a more specific rank (e.g., class, order, ...). Is there an ANI cutoff used to assert that a classification can only be resolved to the phylum level or is an LCA approach across a set of closest SGBs used?

As explained in the answers to the previous questions, whole-genome nucleotide similarity quantification methods show limitations in estimating the genetic distance between genomes that are too phylogenetically distant. Applying the same approach used to define GGBs and FGBs to define class-, order, or phylum-level genome bins is thus unfeasible, as we discussed and backed-up by data in Pasolli et al, Cell, 2019. However, for those SGBs assigned to a GGB and FGB lacking a taxonomic label at genus and family level, respectively, we assign the phylum-level taxonomic label of the closest reference genome regardless of its genetic distance. Because of this, these phylum-level taxonomies should be considered only as a very high-level indication, as they do not imply that a given genome is actually part of that phylum but only that the closest reference belongs to that specific phylum, and this can be a starting point for explorative phylogenetic analysis. We report this limitation now in the Methods as reported above. Based on the useful comments from the previous review, we looked at the set of reference genomes found to be close to the MAGs in the uSGB 19436, and we noticed that the closest is a reference genome assigned to the Chlamydiae phylum, but the others were assigned to the Proteobacteria phylum. Based on this observation, we revised our approach to assigning phylum-level taxonomies as we now first identify the closest reference genome and then select all reference genomes that are within 5% distance from it, then we use a majority voting rule to decide which phylum taxonomic label to report, i.e. the one associated with most of the reference genomes identified as described above. This changed - and corrected! - the phylum label assigned to the uSGB 19436 from Chlamydiae to Proteobacteria and led to the new phylogenetic analysis and improved the overall phylum-level assignment approach in PhyloPhlAn 3.0.

To better explain this point, we modified the text as follows in the “Phylogenetic context for taxonomically assigned and unassigned input genomes” section:

In the Ethiopian study, we focused on the prevalent human gut colonizer *Escherichia coli* (kSGB ID 10068), and on the most prevalent uSGB (ID 19436, 13 MAGs in total) for which the closest reference genomes are assigned to the Proteobacteria phylum.

And in the last part of the “Pipeline for taxonomic assignment of genomes and MAGs” section as follows:

If the average genetic distance of the input genome is >30% to any FGBs, limitations in nucleotide similarity quantification methods would not allow reliable higher-level taxonomic assignment ⁴¹. In these cases, PhyloPhlAn 3.0 reports the phylum label of the set of closest reference genomes (i.e. the set of genomes within 5% genetic distance from the closest) decided via majority voting.

Reviewer #2 (Remarks to the Author):

I am happy to say that the authors did a nice job in addressing the reviewers' comments. Only few and considerably minor points remain.

1. By stating that the data confirms that the ethopian uSGB (ID 19436) represents a new species and genus, the authors either imply that the genus *Campylobacter* is paraphyletic, or the three shown in figure 3C is wrong. Either is possible, and I think this should be briefly discussed. Note in this context that a bootstrap support of 80 is not sufficient evidence that the tree is true. There is a long list of literature discussing how modelling artefacts in the course of phylogeny reconstruction can come to highly supported but wrong trees.

We thank the Reviewer for their comments that helped improve the manuscript and results. We agree that either the *Campylobacter* genus is paraphyletic or that the tree shown in **Figure 3C** could be a modelling artifact from the phylogeny reconstruction. During our analysis, we found that genomes labelled as *Campylobacter* possessed a high genetic diversity (more than 30% ANI distance) that is larger than, for instance, the threshold we use to define FGBs (i.e. family-level phylogenetic diversity), suggesting that either the divergence of this genus or their taxonomic labels should be revised. Moreover, from the phylogenies shown in both **Figure 3C** and **Supplementary Fig. 4**, it is clear that the Ethiopian MAGs, regardless of whether they are diving the *Campylobacter* genus or not, are really distant from any sequenced genomes from the *Campylobacter* genus. We have updated the text to include this discussion:

The 812 publically available genomes in 108 SGBs assigned to distinct species of *Campylobacter*, reveal this genus to be extremely wide encompassing substantially more than 30% genetic distance (ANI analysis in **Supplementary Fig. 4**) which is a diversity usually characterizing whole classes or orders (Pasolli et al. 2019). This suggests that this genus should be revised as also independently confirmed in other taxonomic re-organization efforts (Parks et al. 2018).

Moreover, a quick check showed that both the paraphyletic structure and the diversity of *Campylobacter* are confirmed by independent investigators. In particular, the latest release of the GTDB initiative (Parks et al., 2018) places *Campylobacter* as a new phylum (https://gtdb.ecogenomic.org/searches?q=%25GCA_000175875%25&s=al) and splits it into multiple genera (https://gtdb.ecogenomic.org/searches?q=%25g_Campylobacter%25&s=al). We added this reference in the text as reported above.

We also appreciate the comment of the Reviewer about the bootstrap support of 80, but with the clarification above (including the confirmation in the GTB) and the ANI distance highlighted by **Supplementary Fig. 4**, we think we have strong confirmation for the genus-level diversity of uSGB 19436.

2. I am missing a scale bar in Supplementary Figure 4

We thank the reviewer for pointing out the lack of the scale in the phylogeny in **Supplementary Fig. 4**, we updated the figure.

Supplementary Figure 4: Uncollapsed phylogeny of uSGB 19436 with its closest phyla. Uncollapsed phylogeny of the Ethiopian MAGs assigned to the Proteobacteria phylum (class Epsilonproteobacteria) together with the genomes reconstructed in (Pasolli et al. 2019) assigned to uSGB 19436 and the 589 genomes from the closest phyla: Proteobacteria (Epsilonproteobacteria,

non-monophyletic with the Proteobacteria), Spirochaetes, Chlamydiae, Planctomycetes, *Candidatus* Omintrophica, Lentisphaerae, and Verrucomicrobia. In the top-right inset, genomes are compared using the average nucleotide identity (ANI) measure.

3. I appreciate that the authors added the quartet distance as a measure to compare the trees shown in Suppl Fig 2A and B. I am just wondering to what extent the result of 90.5% consistency between quartet distances the authors report is driven by the many quartets that connect the very closely related strains. If I look at the two phylogenies, I see quite some differences in the topology within the clade making up the right part of the tree. I use the numbers next to the individual sub-clades as labels to clarify my point: Clade 97 is placed as sister to (772,(5,228)) in the curated phylogeny, while it is placed differently in the PhyloPhlAn2 tree. Likewise, the clade 88 is placed differently, as are (25,96). In essence, on a closer look it appears that there quite some differences in the tree topology, and I have the feeling that they are somehow watered down. I suggest to mention these differences and to then explain - if this is true - that the differences are caused by rather poorly supported splits in the trees.

We agree with the reviewer that the different number of genomes for the different *S. aureus* sequence types (STs) can bias the quartet distance computation as several quartets will be among genomes of the same ST, hence very close phylogenetically. To control for this, we randomly chose one genome for each ST and picked the same for the two phylogenies (so as to have no unresolved quartets) and recomputed the quartet distance between the PhyloPhlAn 3.0 tree and the manually curated tree from (Manara et al. 2018) by considering only the previously identified set of leaves, which led to a consistency of 92%. Additionally, we randomly removed fixed numbers of genomes across 100 iterations from both trees and observed a stable quartet distance consistency between both topologies (median ~90.5%, **Response Figure 1**).

Response Figure 1. Quartet distance consistency after randomly removing an increasing number of genomes from the two phylogenies (100 times).

These additional analyses indicate that while there are some notable differences between the two topologies, these don't appear to be watered down by closely related genomes and

could be driven by the set of phylogenetic markers used to build the phylogenies. We've added the single genome per ST quartet distance analysis (in the form of a ternary plot) to **Supplementary Fig. 2** and updated the text as follows:

While some minor differences do occur when comparing PhyloPhlAn 3.0's tree with the previous manually curated phylogeny (e.g. the placement of ST97 with respect to the closest STs), we found an overall correlation of 0.992 (Pearson's correlation) between normalized pairwise branch length distances from the two phylogenies (**Fig. 2B**) and 90.5% (92% when considering one genome for each sequence type [ST]) consistency between quartet distances (**Supplementary Fig. 2**).

Supplementary Figure 2. Side-by-side comparison of *Staphylococcus aureus* phylogenies. (A) The original manually curated ¹ phylogeny. (B) The phylogeny automatically reconstructed by PhyloPhlAn 3.0 and visualized with GraPhlAn using the same annotations of (A). (C) Ternary plot of quartet distances showing the consistency between *S. aureus* phylogenies. (D) Ternary plot of quartet distances of the two *S. aureus* phylogenies reduced to a single genome for each sequence type.

4. I am somehow not really happy with the recurrent use of tree-of-life. For example, what is a 'tree-of-life'-size phylogeny? Moreover, throughout your analysis, you are showing only trees without eukaryotes. Thus, it is only the bacterial and archaeal section of the TOL. I leave it to the discretion of the authors to solve this issue.

We agree with the Reviewer that the use of “tree-of-life” might be misleading. We therefore modified the manuscript in multiple instances to specify that we are referring to “prokaryotic tree-of-life” phylogenies, as reported below.

In the introduction:

It automatically uses species-specific sets of core proteins, stably identified using UniRef90 gene families, to build accurate strain-level phylogenies, while also scaling to tens of thousands of genomes for inferring deep branching, very large size phylogenies.

In the caption of Figure 3:

(C) PhyloPhlAn 3.0 phylogeny of Ethiopian MAGs assigned to uSGB ID 19436 including all reference genomes for the closest phyla (589 in total) according to the prokaryotes tree-of-life in Fig. 4.

In the section “PhyloPhlAn 3.0 can scale to microbial tree-of-life reconstruction including >17,000 genomes and MAGs” we modified the text as follows:

Although phylogenies spanning all the known bacterial and archaeal phyla using more sites and more extensive computation could be used as a default reference (Zhu et al. 2019), the automatic PhyloPhlAn 3.0 pipeline provides a convenient way to incorporate new MAGs and update genome sets.

This is achieved while maintaining high phylogenetic accuracy, as shown by previous clade-specific analyses focusing on organisms from the human microbiome⁴¹, by the overall consistency of the PhyloPhlAn 3.0 tree with the current reference prokaryotic tree-of-life²⁶ (**Supplementary Fig. 6**), and by the comparison of the PhyloPhlAn 3.0 approach of using hundreds of universal markers against other prokaryotic tree-of-life phylogenies based on taxonomy or neighbour-joining (Robinson-Foulds distance <0.3) reported elsewhere²⁶.

And in the Conclusions as follows:

Comparison of automatically obtained phylogenies with respect to manually curated and evaluated phylogenetic trees showed that PhyloPhlAn 3.0 is highly accurate at different resolutions, ranging from species-level clades to the whole prokaryotic tree-of-life.

Ingo Ebersberger

Reviewer #3 (Remarks to the Author):

The paper "Precise phylogenetic analysis of microbial isolates and genomes from metagenomes using PhyloPhlAn2" describes a significant update to the PhyloPhlAn software. This is a review of an initial revision to the paper.

I appreciate the authors' responses to the reviewers, and the revised paper reads quite well. With respect to my previous review, I could not make time to attempt to rerun the software for this revision, my apologies.

I have a few relatively minor suggestions for revision, and one major concern.

My major concern stems from the fact that the NCBI taxonomy is increasingly problematic for classifying genomic content in at least some areas of the taxonomic tree; this is what (I believe) spurred the development of the GTDB taxonomy, which seeks to eliminate these inconsistencies. Given confusion around taxonomic assignments as well as a low but steady rate of contamination in newly incorporated MAGs, it is likely that there will be challenges ahead for any taxonomic software in classification if they use the NCBI taxonomy. So my major concern is this: how does PhyloPhlAn2 deal with confused NCBI taxonomy?

For example, we have found that phyla Actinobacteria and Acidobacteria contain species-level genomes that are nearly identical between the two phyla. The same is true of Actinobacteria and Chloroflexi, and Chlamydiae and Firmicutes, among others.

We also find that GTDB-Tk (which is marker based) and ANI-based approaches yield quite different answers for many MAGs, which highlights the challenges facing the field. (We do not yet understand why; this is ongoing work.)

I note that in the authors' response to Reviewer #1, they highlight a potential taxonomic assignment inconsistency. It might be worth making a point about this in the paper, and or providing a flag for this in the output of PhyloPhlAn2.

It is clearly not reasonable to require PhyloPhlAn2 to fix NCBI's classification, but I do think it is important for PhyloPhlAn2 to alert the user in situations where the NCBI taxonomy may be problematic and inconsistent for the results reported by PhyloPhlAn2.

We agree with the Reviewer that, in general, genome-based taxonomic assignments might be problematic because of the confused NCBI taxonomies. This is why PhyloPhlAn 3.0 uses taxonomic labels based on the taxonomy-independent clustering used to define SGBs (see Pasolli et al, Cell, 2019). In brief, and as now reported in the Methods, the taxonomic label assigned to each specific SGB is based on the majority voting rule on the sets of NCBI taxonomy labels of each reference genome. This approach (more extensively described in Pasolli et al.) solves many taxonomic inconsistencies, for example, those due to genomes from a well-characterized species with completely wrong labels (as they will be assigned the right species-label based on the SGB they belong to) or those labelled "sp." or with a new species name for known species (as the majority of the genomes in the SGB will have the

right species name). This results in a similar taxonomic curation effort for species as in the GTDB; the GTDB additionally re-organizes internal taxonomic labels, whereas the SGB-system focuses on incorporating as many MAGs as possible.

So, while PhyloPhlAn 3.0 does not perform taxonomy curation directly, it uses the taxonomically-curated SGB/GGB/FGB system. We thank the Reviewer for pointing this out and allowing us to highlight these aspects in the revised manuscript. To include the above information in the manuscript, we added the following text in the section “Pipeline for taxonomic assignment of genomes and MAGs” in the Methods:

For the cases in which PhyloPhlAn 3.0 cannot assign an SGB to an input genome, the assignment procedure is repeated at the level of genus-level genome bins (GGBs) and family-level genome bins (FGBs). Similarly to SGBs, GGBs and FGBs were defined elsewhere⁴¹ via hierarchical average linkage clustering at 15% and 30% genetic distance, respectively. These thresholds were empirically estimated in the same work as those more closely reflecting the genetic span of the known taxonomically defined genera and families. GGBs and FGBs are also taxonomically assigned to known genus and family labels if the clusters comprise one or more reference genomes within the corresponding average genetic distance (15% for GGBs, 30% for FGBs, in the case of taxonomic inconsistencies in reference genomes falling inside the same SGB/GGB/FGB, a majority voting approach is applied to assign the most represented taxonomic label). Using this definition of GGBs and FGBs, PhyloPhlAn 3.0 assigns input genomes missing SGB assignment (i.e. the input genome is at >5% average genetic distance with respect to all SGBs) to the closest GGB and/or FGB that are at an average genetic distance <15% and <30%, respectively. If the average genetic distance of the input genome is >30% to any FGBs, limitations in nucleotide similarity quantification methods would not allow reliable higher-level taxonomic assignment⁴¹. In these cases, PhyloPhlAn 3.0 reports the phylum label of the set of closest reference genomes (i.e. the set of genomes within 5% genetic distance from the closest) decided via majority voting.

And the following sentences in the section “Robust taxonomy assignment for metagenome-assembled genomes using species-level genome bins” in the Results:

These span 16,331 SGBs, of which 12,535 have a confident species label based on previous validations alleviating problems of NCBI taxonomic consistency because species labels are assigned to consistently clustered genomes by majority voting (see **Methods**).

Actually, reading through the new text “PhyloPhlAn2 can scale to microbial tree of life reconstructions...” it seems like the database preparation step may simply make arbitrary choices here (based around dereplicating genomes on 95% ANI). If that's the case, then the software may not be able to flag confusing situations, and that should be noted explicitly.

We thank the Reviewer for this comment. The phylogeny presented in Figure 4 contains one genome for each SGB to avoid having many genomes of the same species in the phylogeny. This because the objective is to build a prokaryote tree of life for which strain-level variation is not relevant as well as not to have the phylogeny over-representing such bacterial species that have many (for some, thousands) genomes. The dereplication at 95% ANI is not done by PhyloPhlAn 3.0, but it is a choice we made in this analysis from the PhyloPhlAn 3.0 user's viewpoint. In this analysis, we thus made PhyloPhlAn 3.0 use only one genome for each of the 19,607 SGBs. We thank the Reviewer for raising this point that we now address in the Results as follows:

These were dereplicated prior to PhyloPhlAn application to one representative per species by hierarchical clustering on genomic distances as estimated by Mash (Ondov et al. 2016) with cluster cutoff at 5% intra-cluster nucleotide identity (see Methods), resulting in 19,607 clusters.

Minor suggestions --

Line 223, the Zou paper used EzBioCloud to assign taxonomic identity, it seems. I'm not sure about the use of the word "accuracy" in that sentence; perhaps consistency would be a better word? (This is a soft suggestion, I leave it up to the authors.)

Thank you for the suggestion, we modified the sentence as follows:

The taxonomic labels inferred by PhyloPhlAn 3.0 were also very consistent (97.7%, Supplementary Table 2) with those assigned at species level in the original work highlighting the consistency of the automatic algorithm.

Please provide a version-specific DOI for the software used in this publication.

We thank the Reviewer for pointing out this, we archived the version of PhyloPhlAn 3.0 used in this manuscript on Zenodo to have a DOI: <http://doi.org/10.5281/zenodo.3727181>, and we also updated the "Data availability" section:

PhyloPhlAn 3.0 is released open-source and available in GitHub at <https://github.com/biobakery/phylophlan> and the version used in this work is archived with DOI: <http://doi.org/10.5281/zenodo.3727181>. Manuals and online tutorials describing the PhyloPhlAn 3.0 framework are available at <https://github.com/biobakery/phylophlan/wiki>. User support is provided both through the issues tracking system in the GitHub repository (<https://github.com/biobakery/phylophlan/issues>) and the bioBakery help forum (<https://forum.biobakery.org>). Raw metagenomes for the Ethiopian cohort are available under NCBI-SRA BioProject id PRJNA504891 and the 369 MAGs can be

downloaded from the software page at
<http://segatalab.cibio.unitn.it/tools/phyloplan3>.